# Sex-Dependent Differences in Predictive Value of the C_2_HEST Score in Subjects with COVID-19—A Secondary Analysis of the COLOS Study

**DOI:** 10.3390/v14030628

**Published:** 2022-03-17

**Authors:** Piotr Rola, Adrian Doroszko, Małgorzata Trocha, Katarzyna Giniewicz, Krzysztof Kujawa, Marek Skarupski, Damian Gajecki, Jakub Gawryś, Tomasz Matys, Ewa Szahidewicz-Krupska, Barbara Adamik, Krzysztof Kaliszewski, Katarzyna Kiliś-Pstrusińska, Agnieszka Matera-Witkiewicz, Michał Pomorski, Marcin Protasiewicz, Marcin Madziarski, Urszula Chrostek, Joanna Radzik-Zając, Anna Radlińska, Anna Zaleska, Krzysztof Letachowicz, Wojciech Pisarek, Mateusz Barycki, Janusz Sokołowski, Ewa Anita Jankowska, Katarzyna Madziarska

**Affiliations:** 1Department of Cardiology Provincial Specialized Hospital Iwaszkiewicza 5 Str., 59-220 Legnica, Poland; mateusz.barycki@gmail.com; 2Clinical Departmentof Internal and Occupational Diseases, Hypertension and Clinical Oncology, Wroclaw Medical University, Borowska 213, 50-556 Wroclaw, Poland; adrian.doroszko@umw.edu.pl (A.D.); damian.gajecki@umw.edu.pl (D.G.); jakub.gawrys@umw.edu.pl (J.G.); tomasz.matys@umw.edu.pl (T.M.); ewa.szahidewicz-krupska@umw.edu.pl (E.S.-K.); 3Department of Pharmacology, Wroclaw Medical University, Mikulicz-Radecki Street 2, 50-345 Wroclaw, Poland; malgorzata.trocha@umw.edu.pl; 4Statistical Analysis Centre, Wroclaw Medical University, K. Marcinkowski Street 2-6, 50-368 Wroclaw, Poland; katarzyna.giniewicz@umw.edu.pl (K.G.); krzysztof.kujawa@umw.edu.pl (K.K.); 5Faculty of Pure and Applied Mathematics, Wroclaw University of Science and Technology, Wybrzeże Wyspiańskiego Street 27, 50-370 Wroclaw, Poland; marek.skarupski@pwr.edu.pl; 6Clinical Department of Anaesthesiology and Intensive Therapy, Wroclaw Medical University, Borowska Street 213, 50-556 Wroclaw, Poland; barbara.adamik@umw.edu.pl; 7Department of General, Minimally Invasive and Endocrine Surgery, Wroclaw Medical University, Borowska Street 213, 50-556 Wroclaw, Poland; krzysztof.kaliszewski@umw.edu.pl; 8Clinical Department of Paediatric Nephrology, Wroclaw Medical University, Borowska Street 213, 50-556 Wroclaw, Poland; katarzyna.kilis-pstrusinska@umw.edu.pl; 9Screening of Biological Activity Assays and Collection of Biological Material Laboratory, Wroclaw Medical University Biobank, Wroclaw Medical University, Borowska Street 211A, 50-556 Wroclaw, Poland; agnieszka.matera-witkiewicz@umw.edu.pl; 10Clinical Department of Gynecology and Obstetrics, Wroclaw Medical University, Borowska Street 213, 50-556 Wroclaw, Poland; michal.pomorski@umw.edu.pl; 11Clinical Department and Clinic of Cardiology, Wroclaw Medical University, Borowska Street 213, 50-556 Wroclaw, Poland; marcin.protasiewicz@umw.edu.pl; 12Clinical Department of Rheumatology and Internal Medicine, Wroclaw Medical University, Borowska Street 213, 50-556 Wroclaw, Poland; madziarski.marcin@gmail.com; 13Department of Paediatric Traumatology and Emergency Medicine, Wroclaw Medical University, O. Bujwida Street 44a, 50-345 Wrocław, Poland; urszula.chrostek@umw.edu.pl; 14Clinical Department of Internal Medicine, Pneumology and Allergology, Wroclaw Medical University, M. Skłodowskiej-Curie Street 66, 50-369 Wrocław, Poland; joanna.radzik-zajac@umw.edu.pl (J.R.-Z.); anna.radlinska@umw.edu.pl (A.R.); anna.zaleska@umw.edu.pl (A.Z.); 15Clinical Department of Nephrology and Transplantation Medicine, Wroclaw Medical University, Borowska Street 213, 50-556 Wroclaw, Poland; krzysztof.letachowicz@umw.edu.pl (K.L.); katarzyna.madziarska@umw.edu.pl (K.M.); 16Clinical Department of Gastroenterology and Hepatology, Wroclaw Medical University, Borowska Street 213, 50-556 Wroclaw, Poland; wojciech.pisarek@umw.edu.pl; 17Department of Emergency Medicine, Wroclaw Medical University, Borowska Street 213, 50-556 Wroclaw, Poland; janusz.sokolowski@umw.edu.pl; 18Institute of Heart Diseases, Wroclaw Medical University, Borowska Street 213, 50-556 Wroclaw, Poland; ewa.jankowska@umw.edu.pl; 19Institute of Heart Diseases, University Hospital in Wroclaw, Borowska Street 213, 50-556 Wroclaw, Poland

**Keywords:** risk factors, COVID-19, SARS-CoV-2, predicting value, mortality, C_2_HEST score, gender differences

## Abstract

*Background:* Since the outbreak of the COVID-19 pandemic, a growing number of evidence suggests that COVID-19 presents sex-dependent differences in clinical course and outcomes. Nevertheless, there is still an unmet need to stratify the risk for poor outcome at the beginning of hospitalization. Since individual C_2_HEST components are similar COVID-19 mortality risk factors, we evaluated sex-related predictive value of the score. *Material and Methods:* A total of 2183 medical records of consecutive patients hospitalized due to confirmed SARS-CoV-2 infections were analyzed. Subjects were assigned to one of two of the study arms (male vs. female) and afterward allocated to different stratum based on the C_2_HEST score result. The measured outcomes included: *in-hospital*-mortality, *three-month-* and *six-month-*all-cause-mortality and *in-hospital* non-fatal adverse clinical events. *Results:* The C_2_HEST score predicted the mortality with better sensitivity in female population regarding the short- and mid-term. Among secondary outcomes, C_2_HEST-score revealed predictive value in both genders for pneumonia, myocardial injury, myocardial infarction, acute heart failure, cardiogenic shock, and acute kidney injury. Additionally in the male cohort, the C_2_HEST value predicted acute liver dysfunction and all-cause bleeding, whereas in the female arm-stroke/TIA and SIRS. *Conclusion:* In the present study, we demonstrated the better C_2_HEST-score predictive value for mortality in women and illustrated sex-dependent differences predicting non-fatal secondary outcomes.

## 1. Introduction

Since the outbreak in 2019 in China of the coronavirus disease (COVID-19), the pandemic has revealed an unprecedented impact on the global health care system, with over 450 million confirmed cases resulting in approximately 6 million of deaths reported worldwide [1]. From the initial phase of the pandemic, a growing number of evidence [2] suggests that COVID-19 presents significant sex-dependent differences in clinical course and mortality.

The clinical manifestation of COVID-19 remains unpredictable and varies from asymptomatic to severe or lethal [3,4,5]. Hence, there is an urgent need to introduce a simple and fast triage tool to clinical practice aimed at supporting the decision-making process for the clinicians in terms of appropriate management and optimized use of limited resources.

The C_2_HEST score was originally designed [6] to predict the potential development of atrial fibrillation (AF) in the general population. Lately, a growing body of evidence has appeared, illustrating that the C_2_HEST score can predict poor outcomes of patients in severe clinical conditions. Our previous study demonstrated the usefulness of the C_2_HEST-score in predicting the adverse COVID-19-outcomes in hospitalized subjects with type 2 diabetes mellitus. Since male sex is postulated to be an independent risk factor of an unfavorable COVID-19 outcome, we aimed to assess the sex-dependent predictive value of the C_2_HEST-score.

## 2. Materials and Methods

### 2.1. Study Design and Population

The study population consisted of 2183 consecutive patients with confirmed by reverse transcription-polymerase chain reaction (RT-PCR) infection of SARS-CoV-2 admitted to the Medical University COVID-19 Center. All subjects were hospitalized between February 2020 and June 2021. The study protocol has been approved by the Institutional Review Board and Ethics Committee at the Wroclaw Medical University, Wroclaw, Poland (No: KB-444/2021). All medical data were fully anonymized and retrospectively analyzed. Due to the character of the study protocol written informed consent from participants was not required. Subjects were assigned to one of two of the study arms male vs. female. Subsequently, all patients were assigned into one C_2_HEST score stratum. The C_2_HEST score value was calculated depending on originally proposed variables; coronary artery disease (CAD) (1 point), chronic obstructive pulmonary disease (COPD) (1 point), hypertension (1 point), elderly (age ≥ 75 years, 2 points), systolic heart failure (HF) (2 points), and thyroid disease (1 point). Based on the calculated score subjects were allocated to one of three stratum -*low-risk* 0 or 1 point, *medium-risk* 2 or 3 points, and *high-risk 4* and more points.

### 2.2. Follow-Up and Outcomes

The primary clinical outcome was an *in-hospital*, *three-month-*, and *six-month-*all-cause mortality. Other clinical outcomes focused on *in-hospital*: end of hospitalization other than death (discharge, deterioration or recovery with subsequent transfer to another hospital) advanced mechanical ventilation support, shock, multiple organ dysfunction syndrome (MODS), systemic inflammatory response syndrome (SIRS), sepsis. Also, other clinical features were collected symptomatic bleeding, pneumonia, pulmonary embolism, acute heart failure, myocardial injury, stroke, acute kidney injury, acute liver dysfunction. 

### 2.3. Statistical Analysis 

Statisticians with experience in medical academic research performed the analyses to this manuscript. The R language version 4.0.4 with additional packages-pROC and time-ROC [7], survival [8], coin [9], and odds ratio was used for the purpose of data analysis [10] A level of 0.05 was set as significance value. 

Descriptive data regarding categorical variables are shown as numbers and percentages, whereas for numerical variables as mean with standard deviation, range (minimum-maximum) along with the number of non-missing values. The omnibus and chi-square tests were performed for categorical variables which exceeded five expected cases in each group. The Fisher exact test was performed for subjects with fewer cell counts. The Welch’s ANOVA was set up for continuous variables in order to adjust for unequal variances between the risk-strata and sample size large sufficient for appropriateness of asymptotic results. For continuous variables, the Games-Howell’s variant of Tukey correction was performed as a part of a post-hoc analysis. On the other hand, for categorical variables, the post-hoc test was analogous to the omnibus test. However, it was performed in subgroups with a Bonferroni correction. Due to a fact that the in-hospital mortality along with the all-cause mortality were available as right-censored data, the time-dependent ROC analysis with inverse probability of censoring weighting (IPCW) was used to estimate them. The time-dependent area under the curve (AUC) was used to assess the C2HEST score and additionally a confirmation of differences in survival curves among risk strata was obtained by a Log-rank test. Proportional hazard assumption was verified using the Grambsch-Therneau test. During analysis of the hazard ratio (HR) in the C2HEST score, its components, as well as risk strata, a Cox proportional hazard model was used. Dichotomic nature of secondary outcomes resulted in the use of a logistic regression model during their analysis. In order to assess predictive capability, the classical receiver operating characteristic (ROC) analysis with an AUC measure was performed. Odds ratio (OR) was presented as a size effect for the influence of the C2HEST score, its components and risk strata.

## 3. Results

### 3.1. Baseline Demographical and Clinical Features of the Studied Population

The study population was composed of 2183 subjects at mean age 60.1 ±18.8 [17–100] A total of 1101 women at mean age 59.3 ± 21.1 [17–100] were enrolled to this study, who were subsequently assigned to the low-risk *n* = 682 subjects, medium-risk *n* = 284 patients, and high-risk *n* = 135 C_2_HEST strata, respectively. Simultaneously, a total of 1082 males at mean age of 60.8 ± 16.1 [17–99], were assigned to the low-risk (*n* = 735), medium-risk (*n* = 208) and high-risk(*n* = 139). The baseline clinical data of both study cohorts is presented in Table 1. In both cohorts, higher C_2_HEST risk was related to a higher number of comorbidities and more advanced age. 

Data regarding the relationship between the C_2_HEST score result and treatment applied before hospitalization is shown in the Table 2. In the both cohorts along with increased C_2_HEST score, we observed an increasing prevalence drug commonly used in cardiovascular disorders such as angiotensin-converting-enzyme inhibitors (ACEI), mineralocorticoid receptor antagonists (MRA), b-blockers, calcium channel blockers, diuretics, statins, vitamin K antagonists (VKA), novel oral anticoagulants (NOAC), acetylsalicylic acid, P2Y12 inhibitor, metformin, and insulin. 

Table 3 shows the sex-specific baseline characteristics of patient-reported symptoms, and vital signs during the hospital admission in the studied cohort. The female but not male cohort, had significant differences between the C_2_HEST strata regarding the prevalence of cough, smell dysfunction, body temperature, and systolic blood pressure, which were decreasing as the score raised. Opposite findings were observed regarding dyspnoea, heart rate, and the diastolic blood pressure.

The detailed characteristics of the laboratory parameters measured during the hospitalisation in the study cohort were pooled in Table 4 and Table 5.

Both genders revealed significant differences between the C_2_HEST strata and complete blood count parameters along with ion parameters. Noteworthy, no significant differences between strata in terms of initial inflammatory markers (procalcitonin, IL-6, CRP) along with acid-base balance parameters were noted.

The parameters of kidney function, including urea, creatinine, eGFR maintained significantly worse in the high-risk C_2_HEST stratum for both genders, however baseline serum concentration of protein and albumin was significantly lower only in females with higher C_2_HEST score value. In both study cohorts we observed increasing level of cardiac injury markers including troponin T and NT-pro-BNP levels in patientsallocated higher-risk group depending on their C_2_HEST score value. Surprisingly, lipid disorders (level of LDL and triglycerides) noticed at the time of admission were less severe subjects from high-risk stratum in both study cohorts. 

### 3.2. Specific Treatment Applied during Hospitalization

Differences in applied treatment during hospitalization between the C_2_HEST group among genders are highlighted in Table 6. Women in the higher C_2_HEST stratum were prone to receive convalescent plasma. We did not observe any differences among the male cohort. In both study arms, we observed changes in the prevalence of antibiotic application. Subjects from the high-risk stratum more often received this type of therapy.

The assignment to specific C_2_HEST stratum score correlated with the type of respiratory support applied during the hospitalization. Additionally, in the male cohort, it correlated with the prevalence of coronary revascularization procedures during index hospitalization along with the need for the catecholamine’s administration (Table 7).

### 3.3. Association C_2_HEST Score with Results and Mortality 

In the female cohort, the *in-hospital* and *three-month* and*six-month* mortality rates were the highest in *high-risk* C_2_HEST stratum reaching 31.9%, 48.1%, and 61.4%. Noteworthy, mortality rates in the medium-risk stratum were significantly higher than in *low-risk*. All data regarding short and long-term mortality were presented in Table 8. Similarly, in the males’ cohort *in-hospital, three-month* and *six-month* mortality was also highest in the *high-risk* C_2_HEST stratum and come to 38.8%, 59.0%, and 68.8%. Also, in this study arm differences between all C_2_HEST groups were statistically significant.

### 3.4. The All-Cause Mortality Discriminatory Performance of the C_2_HEST Score

The time dependent receiver operating characteristic (ROC) analysis in both study cohorts revealed that the C_2_HEST scale is more sensitive in the female cohort (Figure 1). The C_2_HEST predicting AUC in women vs. man cohorts were higher at all calculated periods. Following the 1-month AUC = 72.5 vs. 70.3% 3-month AUC = 74.6 vs. 71.3%, six-month AUC = 73.8 vs. 68.4 %. All of the data were calculated for all-cause death without competing risk Figure 2 present ROC analysis in the male population. Figure 3 presented the time-dependent AUC for the C_2_HEST score in predicting the all-cause deaths in both cohort, slightly higher AUC value was observed in the female arm. The survival curves for the C_2_HEST stratum in both study cohorts were estimated using Kaplan-Meier functions. The p value for Log-rank test was <0.0001 (Figure 4). We have observed differences in estimated survival probability in both study cohorts. Practically, starting from admission time, the females were more likely to survive the COVID-19. Estimated six-month survival probability for *high-risk* subjects reached 0.5 in the female cohort, while for the male subject was below 0.4. Similarly, in medium-risk-stratum for women the survival probability was above 0.6 when compared to 0.5 in men. Additionally, the *low-risk* subjects in the female cohort maintained at the level of more than 0.9 for the whole observation period while in men reached 0.8, respectively.

Subsequently, two Cox models were analyzed to assess the effect of the C_2_HEST score stratification on COVID-19 mortality. The overall model takes an uncategorized value of the C_2_HEST score, and it met the hazard proportional assumption in both study cohorts. An additional point in the C_2_HEST score resulted in increased the total-death intensity approximately in 42.8% in female subjects (HR 1.428, 95% CI 1.349–1.513 *p* < 0.0001) and respectively in male population 40.0% (HR 1.400, 95% CI 1.331–1.474 *p* < 0.0001). Furthermore, considering the categorized model, the change from the low to the medium category in the female population increased death expectation 4.267 times, and respectively; 3.289 times for males. Subsequently, transfer between the *low-risk* stratum to *high-risk* stratum raised all-cause death intensity 6.52 (female) and 4.476(male) times. The data are shown in Table 9 and Table 10.

The associations of individual C_2_HEST score components with mortality in both study cohorts are presented in Table 11 and Table 12. The highest prognostic value for all-cause- death in both study groups was noticed for age (in women 2.750 vs. 3.059 in men, respectively). Interestingly, coronary artery disease was associated with higher HR for death only in men, whereas the COPD and hypertension only in woman.

Additionally, we verified whether the original cut-off values for particular C_2_HEST score risk (the low/medium/high-risk categories for 0–1/2–3/≥4 points, respectively) is potentially the best possible stratification system. Regarding the difference in Kaplan-Meier survival curves, all of the possible C_2_HEST intervals were analyzed in both study cohorts, and for each, we calculated the log-rank statistics (Table 13 and Table 14). The highest value of log-rank test statistics, presenting the best cut-off point for high (h) and medium (m) strata was obtained for the original C_2_HEST-score risk strata in the female population (m2 and h4, respectively). On the other hand, in male cohort the highest value of the Log-rank corresponded with m2 and h5, which reflects the following strata: 0–1 low, 2–4 medium, 5–8 high. 

### 3.5. Relationship of C_2_HEST Score with Non-Fatal Outcomes

Clinical non-fatal events in the C_2_HEST risk strata in both study arms are presented in Table 15. In both study cohorts, the subjects assigned to the C_2_HEST *high-risk* stratum were characterized by greater prevalence of pneumonia, acute kidney injury, and cardiovascular disorders during hospitalization. This observation regards myocardial injury, myocardial infarction, acute heart failure, and cardiogenic shock. Additional, female subjects with higher C_2_HEST values were more prone to subject a new episode of stroke/transient ischemic attack (TIA), and systemic inflammatory response syndrome (SIRS) during hospitalization. On the other hand, a high C_2_HEST score in the male subpopulation was associated with a higher probability of shock, acute liver dysfunction, and bleeding occurrence.

Additionally, the overall odds ratio for the discriminatory performance of the C_2_HEST score on the clinical non-fatal events was summarized in Figure 5 (female) and Figure 6 (male). Noteworthy, the highest predictive of C_2_HEST score value in the female cohort was achieved for, acute heart failure (OR_overall_ = 2.180, 95%CI 1.778–2.724, *p* = 0.0034). Similar findings were observed in the male cohort -the highest value was observed for acute heart failure (OR_overall_ = 1.861, 95%CI 1.574–2.229, *p* < 0.0001).

## 4. Discussion

Several studies demonstrated [11] no significant differences regarding the susceptibility to the SARS-CoV-2 infection between biological genders. Nevertheless, male gender is an independent risk factor for the poor outcome of COVID-19 including higher severity and fatality rates [12]. Various biological factors may play a role in sex-dependent different responses to the severe acute respiratory syndrome coronavirus 2 (SARS-CoV-2). Biological sex affects the initial phase of infection mainly by sex-based differences in the expression of the ACE2 receptor responsible for the entry of the SARS-CoV-2 into the cells [13]. Sex differences affect also an immune response to viral infection. Females tend to have a lower potency to develop an uncontrolled inflammatory response process [14] with coexisting decreased viral load during the infection. The physiological mechanism of this process is multifactorial [15,16] and includes the sex-specific transcriptional regulatory network, various gen variants especially connected with chromosome X, epigenetic modifications, transcription factors, and sex steroids. Noteworthy, different social, behavioral, and comorbid factors are also postulated [17] to worsen the prognosis in men.

The previously observed sex-dependent dichotomy in the COVID-19 mortality was also confirmed in our study. For all of the three C_2_HEST strata, greater fatality rate in the male cohort compared to the female one was noted. Independently, we confirmed the previously reported usefulness of the C_2_HEST score in predicting the adverse COVID-19 outcomes, including the mortality in both genders. However, despite lower mortality observed in women, the ROC analysis revealed that the C_2_HEST-score is a more sensitive tool in women regarding the short- and mid-term (up to 6 month-) mortality (for 1-month the AUC = 72.5 vs. 70.3%and for 6-month AUC = 73.8 vs. 68.4 % in men, respectively). Gender is often considered among the variables defining the probability of a severe clinical outcome of infection. 

Analysis of individual C_2_HEST score variables in both cohorts revealed differences between gender in features significantly affecting mortality. Beyond age and previously diagnosed heart failure common for both sexes, in the female group, only hypertension and COPD reached statistical significance. On the other hand, in the male cohort such observation was made for coronary artery disease. Although the pathophysiology underlying severe COVID-19 course remains not fully understood, it can be hypothesized that endothelial dysfunction induced by hypertension [18] might abolish the initial favorable female immune response [14] to SARS-CoV-2 infection. Moreover, the endothelial dysfunction promotes microvascular thrombi and pro-thrombotic state associated with respiratory failure and fatal outcome in COVID-19 [19]. On the other hand, the increased mortality rate of COVID-19 male patients with CAD is probably related to the presence of multiple comorbidities [20] or direct myocardial injury connected with enhanced platelet activation induced by SARS-CoV-2 infection [21]. 

It is noteworthy that, besides observed in both genders significant differences in mortality between the C_2_HEST strata, a similar relationship was noticed in the prevalence of pneumonia and cardiovascular non-fatal secondary outcomes (myocardial infarction, myocardial injury, acute heart failure, cardiogenic shock, and acute kidney failure). Our study revealed that in the male cohort alongside with higher C_2_HEST stratum, a greater rate of acute liver injury (ALI), bleedings and shock was present. This observation supports the previously described relationships between male gender and liver impairment in COVID subjects [22]. Although the mechanism of liver injury in SARS-CoV-2 infection remains unclear, a combination of direct viral inclusion of hepatocytes, as well as the result of uncontrolled immune, may be responsible for the damage, which interestingly, have also been associated with poor outcomes in COVID patients [23]. 

Furthermore, some data [5,24] suggests that individuals with gastrointestinal problems particularly those with earlier stages of liver impairment are more prone to develop severe COVID-19 disease with advanced respiratory failure. Concerning epidemiological data a higher prevalence of liver disorders [25] with coexisting higher susceptibility for endothelial dysfunction [26,27] may be important factors affecting outcomes in the male population. 

It is possible that acute liver injury in the male cohort may be also partially responsible for the higher rate of bleedings as a result of coagulation systems disorders (mild elevations of INR, activated partial thromboplastin time (APTT), and thrombin time (TT)) observed in patients with ALI in course of COVID-19 [23,28]. Initial higher level of INR in males high-risk C_2_HEST score stratum seems to support this thesis. Although the principal clinical manifestation of severe COVID-19 is a respiratory failure with a coexisting uncontrolled immune reaction, subjects with COVID-19 show a high incidence of thromboembolic events [29], particularly in fatal cases [30], however antithrombotic treatment prior to COVID-19 infection is unlikely to have a protective effect [31]. Bleeding complications in subjects with COVID-19 give rise to justifiable concerns [32,33] and should always be considered before applying anticoagulation in patients with SARS-CoV-2 infection. Therefore several predictive scores [34] focused on identifying patients at increased risk for major bleeding have been recently proposed. Results of our study suggest that the C2HEST score might be also useful in the identification of the “high-risk for bleeding” subpopulations. However, subsequent studies are needed to define predictive value of the C2HEST score in terms of bleedings.

### Limitations

We have observed several limitations of this study including the retrospective, single-center, character. These factors could affect the validity of our conclusions. Additionally, the study population was homogeneous and consisted of hospitalized patients and not involved ambulatory subjects. Furthermore, all hospitalizations were carried out in the face of limited resources (global COVID-19 pandemic) probably these extraordinary circumstances could partially affect the clinical outcomes.

## 5. Conclusions

To the best of our knowledge, this study is the first demonstration of the sex-dependent differences in the predictive value of the C_2_HEST score in subjects admitted to hospital due to SARS-CoV-2 infection. This simple risk score evaluated during the hospital admission could predict adverse outcomes in both including in-hospital and six-month-mortality and other clinical events such as acute kidney injury, myocardial injury acute heart failure, myocardial infarction, and cardiogenic shock. Additionally in the male cohort, it well correlated with acute liver injury and prevalence of all kinds of bleeding. The simplicity of this scale allows assuming that C_2_HEST-score might become a useful triage tool for risk stratification in both genders with COVID-19.

## Figures and Tables

**Figure 1 viruses-14-00628-f001:**
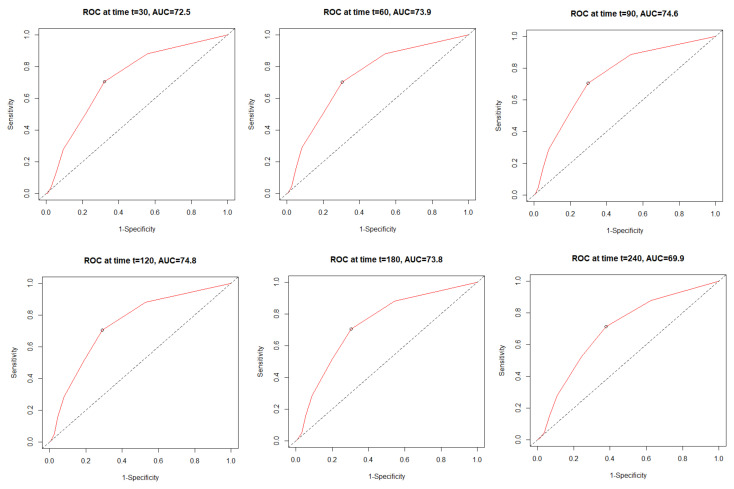
The time dependent receiver operating characteristic (ROC) for all-cause mortality in female cohort.

**Figure 2 viruses-14-00628-f002:**
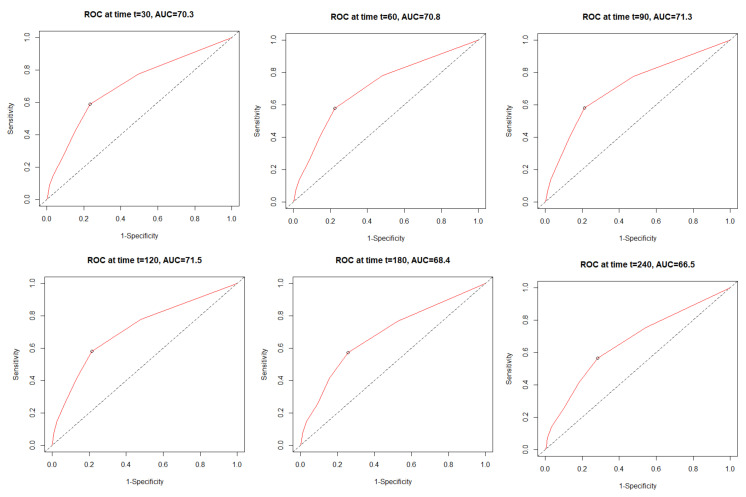
The time dependent receiver operating characteristic (ROC) for all-cause mortality in male cohort.

**Figure 3 viruses-14-00628-f003:**
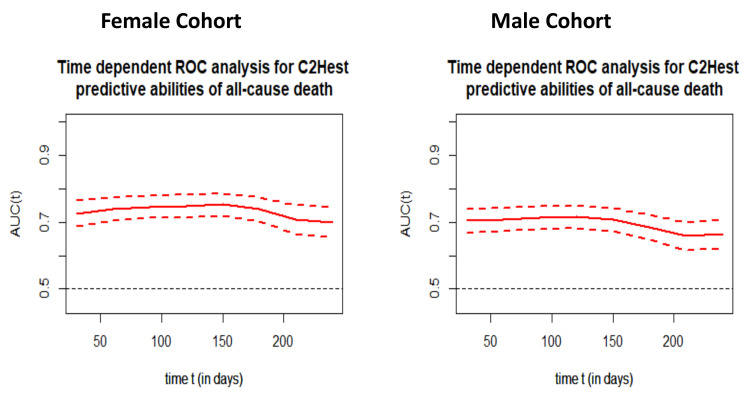
Time-dependent ROC analysis for the C_2_HEST predictive abilities of all cause death in both study cohorts.

**Figure 4 viruses-14-00628-f004:**
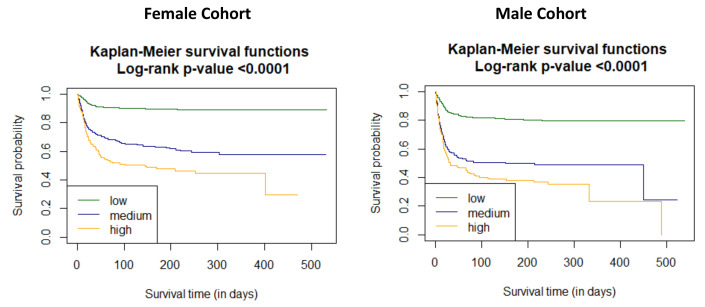
The survival curves for the C_2_HEST stratum in both study cohorts estimated by Kaplan-Meier function.

**Figure 5 viruses-14-00628-f005:**
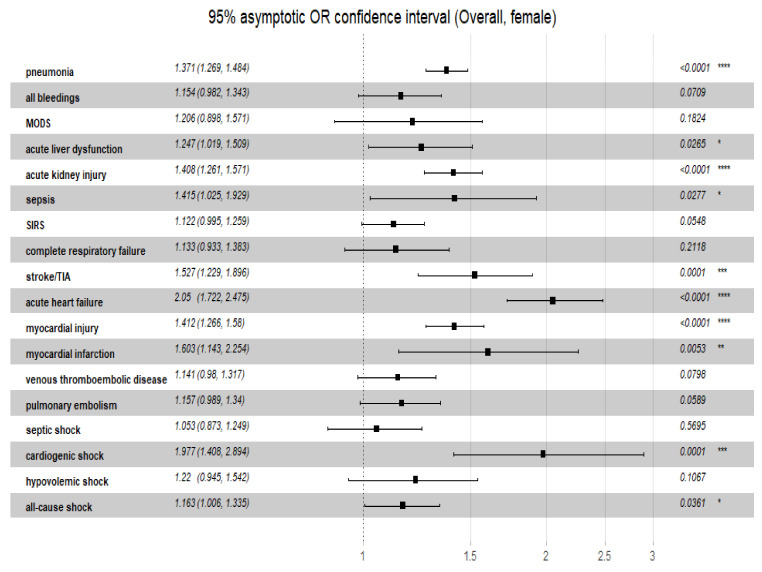
The overall odds ratio for the discriminatory performance of the C_2_HEST score on the clinical non-fatal events in female cohort. Abbreviations: MODS, multiple organ dysfunction syndrome; TIA, transient ischemic attack; SIRS, systemic inflammatory response syndrome. Significance code: * <0.05; ** <0.01; *** <0.001; **** <0.0001.

**Figure 6 viruses-14-00628-f006:**
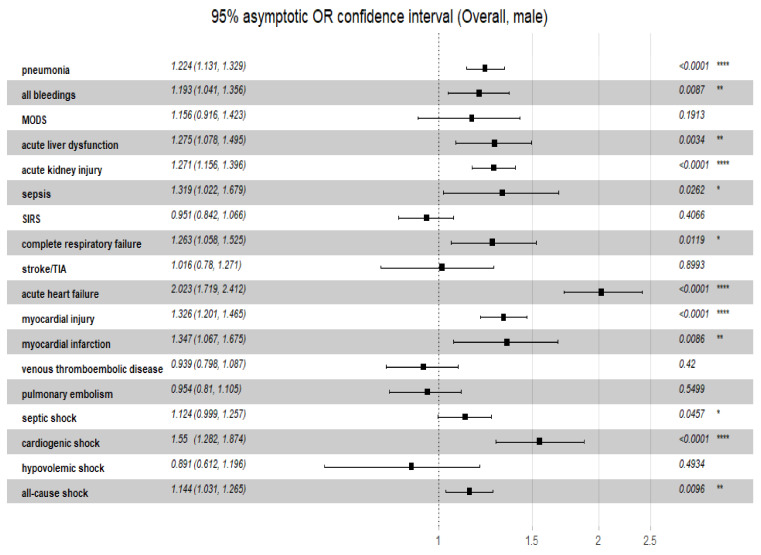
The overall odds ratio for the discriminatory performance of the C_2_HEST score on the clinical non-fatal events in female cohort. Abbreviations: MODS, multiple organ dysfunction syndrome; TIA, transient ischemic attack; SIRS, systemic inflammatory response syndrome. Significance code: * <0.05; ** <0.01; *** <0.001; **** <0.0001.

**Table 1 viruses-14-00628-t001:** Baseline demographics and clinical characteristics.

Variables Units	Low Risk [0–1]		Medium [2–3]		High Risk [≥4]		OMNIBUS *p*-Value		*p* Value for Post-Hoc Analysis	
	Females N = 682	Males N = 735	Females N = 284	Males N = 208	Females N = 135	Males N = 139	Females	Males	Females	Males
Demographics										
**Age, years** mean ± SD/min-max	47.8 ± 17.1 17–74	54.2 ± 14.0 17–74	76.7 ± 12.0 29–100	74.0 ± 1.2 37–99	81.0 ± 8.7 47–100	76.2 ± 9.4 38–92	<0.0001	<0.0001	0.0 ^a,b^ 0.0001 ^c^	0.0 ^a^ <0.0001 ^b^ 0.115 ^c^
**Age ≥ 65 years** *n*/*n*(%)	165 (24.2)	211 (28.7)	247 (87.0)	172 (82.7)	129 (95.6)	123 (88.5)	<0.0001	<0.0001	<0.0001 ^a,b^ 0.0339 ^c^	<0.0001 ^a,b^ 0.5515 ^c^
**BMI, kg/m^2^** mean ± SD/min-max/N	28.3 ±5.3 17.1–45.7 199	28.2 ± 4.8 15.4–49.4 198	30.1 ±5.9 18.6–47.8 48	28.3 ±5.2 20.9–46.7 42	27.1 ±6.7 16.4–45.8 17	28.0 ± 5.6 17.3–48.2 50	0.1255	0.9609	N/A	N/A
**Co-morbidities**										
**Hypertension** *n*/*n*(%)	179 (26.2)	236 (32.1)	213 (75.0)	144 (69.2)	126 (93.3)	123 (88.5)	<0.0001	<0.0001	<0.0001 ^a,b,c^	<0.0001 ^a,b^ 0.0002 ^c^
**Dyslipidaemia** *n*/*n*(%)/N	74 (59.2) 125	138 (57.3) 241	37 (44.6) 83	32 (39.0) 82	29 (48.3) 60	17 (29.8) 57	0.0932	0.00011	N/A	0.0191 ^a^ 0.001 ^b^ 1.0 ^c^
**Atrial fibrilation/flutter** *n*/*n*(%)	14 (2.1)	35 (4.8)	60 (21.1)	46 (22.1)	65 (48.1)	70 (50.4)	<0.0001	<0.0001	<0.0001 ^a,b,c^	<0.0001 ^a,b,c^
**Previous coronary revascularisation** *n*/*n*(%)	0 (0.0)	6 (0.8)	9 (3.2)	28 (13.5)	35 (25.9)	76 (54.7)	<0.0001	<0.0001	<0.0001 ^a,b,c^	<0.0001 ^a,b,c^
**Previous myocardial infarction** *n*/*n*(%)	1 (0.1)	10 (1.4)	18 (6.3)	45 (21.6)	37 (27.4)	80 (57.6)	<0.0001	<0.0001	<0.0001 ^a,b,c^	<0.0001 ^a,b,c^
**Heart failure** *n*/*n*(%)	0 (0.0)	0 (0.0)	20 (7.0)	33 (15.9)	91 (67.4)	111 (79.9)	<0.0001	<0.0001	<0.0001 ^a,b,c^	<0.0001 ^a,b,c^
**Moderate/severe valvular heart disease or previous valve heart surgery** *n*/*n*(%)	7 (1.0)	6 (0.8)	14 (4.9)	18 (8.7)	26 (19.3)	25 (18.0)	<0.0001	<0.0001	0.0012 ^a^ <0.0001 ^b,c^	<0.0001 ^a,b^ 0.0467 ^c^
**Peripheral artery disease** *n*/*n*(%)	7 (1.0)	19 (2.6)	14 (4.9)	17 (8.2)	11 (8.1)	32 (23.0)	<0.0001	<0.0001	0.0012 ^a^ <0.0001 ^b^ 0.5813 ^c^	0.0014 ^a^ <0.0001 ^b^ 0.0006 ^c^
**Previous stroke/TIA** *n*/*n*(%)	17 (2.5)	30 (4.1)	33 (11.6)	26 (12.5)	24 (17.8)	34 (24.5)	<0.0001	<0.0001	<0.0001 ^a,b^ 0.3522 ^c^	<0.0001 ^a,b^ 0.0183 ^c^
**Chronic kidney disease** *n*/*n*(%)	33 (4.8)	37 (5.0)	26 (9.2)	44 (21.2)	39 (28.9)	52 (37.4)	<0.0001	<0.0001	0.0486 ^a^ <0.0001 ^b,c^	<0.0001 ^a,b^ 0.0042 ^c^
**Haemodialysis** *n*/*n*(%)	11 (1.6)	8 (1.1)	5 (1.8)	15 (7.2)	8 (5.9)	11 (7.9)	0.01467	<0.0001	1.0 ^a^ 0.0204 ^b^ 0.0963 ^c^	<0.0001 ^a,b^ 1.0 ^c^
**Asthma** *n*/*n*(%)	32 (4.7)	22 (3.0)	17 (6.0)	3 (1.4)	7 (5.2)	4 (2.9)	0.7053	0.4996	N/A	N/A
**COPD** *n*/*n*(%)	1 (0.1)	5 (0.7)	9 (3.2)	16 (7.7)	16 (11.9)	28 (20.1)	<0.0001	<0.0001	0.0003 ^a^ <0.0001 ^b^ 0.0041 ^c^	<0.0001 ^a,b^ 0.0035 ^c^
**Hypothyroidism** *n*/*n*(%)	65 (9.5)	11 (1.5)	56 (19.7)	12 (5.8)	52 (38.5)	12 (8.6)	<0.0001	<0.0001	<0.0001 ^a,b^ 0.0002 ^c^	0.004 ^a^ <0.0001 ^b^ 1.0 ^c^
**Hyperthyroidism** *n*/*n*(%)	3 (0.4)	1 (0.1)	7 (2.5)	3 (1.4)	3 (2.2)	4 (2.9)	0.0083	0.0009	0.0272 ^a^ 0.1807 ^b^ 1.0 ^c^	0.1065 ^a^ 0.0081 ^b^ 1.0 ^c^

Continuous variables are presented as: mean ± SD, range (minimum–maximum) and number of non-missing values. Categorized variables are presented as: a number with a percentage. Information about the numbers with valid values is provided in the left column. Abbreviations: N, valid measurements; *n*, number of patients with parameter above cut-off point; SD, standard deviation; BMI, body mass index; TIA, transient ischemic attack; COPD, chronic obstructive pulmonary disease; OMNIBUS, analysis of variance; N/A, non-applicable; ^a^ low risk vs. medium risk, ^b^ low risk vs. high risk, ^c^ medium risk vs. high risk. Red color text = statistically significant values.

**Table 2 viruses-14-00628-t002:** Baseline characteristics of the study cohort-treatment applied before hospitalization.

Variables Units	Low Risk [0–1]		Medium [2–3]		High Risk [≥4]		OMNIBUS *p*-Value		*p* Value for Post-Hoc Analysis	
	Females N = 682	Males N = 735	Females N = 284	Males N = 208	Females N = 135	Males N = 139	Females	Males	Females	Males
Treatment applied before hospitalization										
**ACEI** *n*/*n*(%)	47 (6.9)	69 (9.4)	57 (20.1)	63 (30.3)	54 (40.0)	62 (44.6)	<0.0001	<0.0001	<0.0001 ^a,b,c^	<0.0001 ^a,b^ 0.0273 ^c^
**ARB** *n*/*n*(%)	33 (4.8)	43 (5.9)	26 (9.2)	12 (5.8)	14 (10.4)	16 (11.5)	0.0087	0.0413	0.04855 ^a^ 0.0611 ^b^ 1.0 ^c^	1.0 ^a^ 0.0724 ^b^ 0.2546 ^c^
**MRA** *n*/*n*(%)	3 (0.4)	15 (2.0)	13 (4.6)	20 (9.6)	20 (14.8)	29 (20.9)	<0.0001	<0.0001	<0.0001 ^a,b^ 0.0021 ^c^	<0.0001 ^a,b^ 0.0158 ^c^
**β-blocker** *n*/*n*(%)	78 (11.4)	119 (16.2)	102 (35.9)	77 (37.0)	76 (56.3)	81 (58.3)	<0.0001	<0.0001	<0.0001 ^a,b^ 0.0004 ^c^	<0.0001 ^a,b^ 0.0005 ^c^
**Calcium channel blocker dihydropiridines** *n*/*n*(%)	37 (5.4)	66 (9.0)	48 (16.9)	36 (17.3)	34 (25.2)	40 (28.8)	<0.0001	<0.0001	<0.0001 ^a,b^ 0.1863 ^c^	0.003 ^a^ <0.0001 ^b^ 0.0493 ^c^
**α-adrenergic blocker** *n*/*n*(%)	10 (1.5)	35 (4.8)	6 (2.1)	28 (13.5)	8 (5.9)	31 (22.3)	0.0113	<0.0001	1.0 ^a^ 0.0137 ^b^ 0.2272 ^c^	<0.0001 ^a,b^ 0.1358 ^c^
**Amiodarone** *n*/*n*(%)	1 (0.1)	0 (0.0)	1 (0.4)	1 (0.5)	0 (0.0)	1 (0.7)	0.6165	0.1027	N/A	N/A
**Thiazide or thiazide-like diuretic** *n*/*n*(%)	29 (4.3)	39 (5.3)	36 (12.7)	11 (5.3)	16 (11.9)	19 (13.7)	<0.0001	0.0008	<0.0001 ^a^ 0.0026 ^b^ 1 ^c^	1.0 ^a^ 0.0017 ^b^ 0.0345 ^c^
**Loop diuretic** *n*/*n*(%)	13 (1.9)	26 (3.5)	25 (8.8)	40 (19.2)	33 (24.4)	48 (34.5)	<0.0001	<0.0001	<0.0001 ^a,b,c^	<0.0001 ^a,b^ 0.0061 ^c^
**Statin** *n*/*n*(%)	40 (5.9)	63 (8.6)	56 (19.7)	65 (31.3)	49 (36.3)	77 (55.4)	<0.0001	<0.0001	<0.0001 ^a,b^ 0.0012 ^c^	<0.0001 ^a, b, c^
**Acetylsalicylic acid** *n*/*n*(%)	35 (5.1)	46 (6.3)	44 (15.5)	51 (24.5)	33 (24.4)	49 (35.3)	<0.0001	<0.0001	<0.0001 ^a,b^ 0.1137 ^c^	<0.0001 ^a,b^ 0.1234 ^c^
**The second antiplatelet drug** *n*/*n*(%)	1 (0.1)	6 (0.8)	5 (1.8)	5 (2.4)	4 (3.0)	18 (12.9)	0.0009	<0.0001	0.0292 ^a^ 0.0094 ^b^ 1.0 ^c^	0.2154 ^a^ <0.0001 ^b^ 0.0007 ^c^
**LMWH** *n*/*n*(%)	32 (4.7)	42 (5.7)	23 (8.1)	18 (8.7)	11 (8.1)	15 (10.8)	0.0674	0.0535	N/A	N/A
**VKA** *n*/*n*(%)	4 (0.6)	6 (0.8)	6 (2.1)	8 (3.8)	10 (7.4)	13 (9.4)	<0.0001	<0.0001	0.2172 ^a^ <0.0001 ^b^ 0.038 ^c^	0.0129 ^a^ <0.0001 ^b^ 0.1213 ^c^
**NOAC** *n*/*n*(%)	6 (0.9)	12 (1.6)	22 (7.7)	15 (7.2)	23 (17.0)	29 (20.9)	<0.0001	<0.0001	<0.0001 ^a,b^ 0.0207 ^c^	0.0002 ^a^ <0.0001 ^b^ 0.001 ^c^
**Insulin** *n*/*n*(%)	23 (3.4)	39 (5.3)	14 (4.9)	15 (7.2)	22 (16.3)	18 (12.9)	<0.0001	0.0038	1.0 ^a^ <0.0001 ^b^ 0.0007 ^c^	1.0 ^a^ 0.0047 ^b^ 0.3296 ^c^
**Metformin** *n*/*n*(%)	40 (5.9)	64 (8.7)	35 (12.3)	32 (15.4)	22 (16.3)	29 (20.9)	<0.0001	<0.0001	0.0031 ^a^ 0.0002 ^b^ 1.0 ^c^	0.022 ^a^ 0.0001 ^b^ 0.7261 ^c^
**SGLT2 inhibitor** *n*/*n*(%)	4 (0.6)	7 (1.0)	4 (1.4)	3 (1.4)	3 (2.2)	6 (4.3)	0.12658	0.018	N/A	1.0 ^a^ 0.0286 ^b^ 0.4938 ^c^
**Oral antidiabetics other than SGLT2 inhibitor and metformin** *n*/*n*(%)	10 (1.5)	17 (2.3)	20 (7.0)	14 (6.7)	11 (8.1)	17 (12.2)	<0.0001	<0.0001	<0.0001 ^a,b^ 1.0 ^c^	0.01 ^a^ <0.0001 ^b^ 0.3507 ^c^
**Proton pump inhibitor** *n*/*n*(%)	31 (4.5)	58 (7.9)	39 (13.7)	36 (17.3)	37 (27.4)	49 (35.3)	<0.0001	<0.0001	<0.0001 ^a,b^ 0.0034 ^c^	0.0003 ^a^ <0.0001 ^b^ 0.0007 ^c^
**Oral corticosteroid** *n*/*n*(%)	31 (4.5)	31 (4.2)	17 (6.0)	7 (3.4)	5 (3.7)	1 (0.7)	0.5164	0.125	N/A	N/A
**Immuno-suppression** **other than oral corticosteroid** *n*/*n*(%)	24 (3.5)	25 (3.4)	12 (4.2)	10 (4.8)	2 (1.5)	0 (0.0)	0.3606	0.0185	N/A	1.0 ^a^ 0.0686 ^b^ 0.0209 ^c^

Categorized variables are presented as: a number with a percentage. Information about the numbers with valid values is provided in the left column. Abbreviations: N, valid measurements; *n*, number of patients with parameter above the cut-off point; ACEI, angiotensin-converting-enzyme inhibitors; ARBs, angiotensin receptor blockers; MRAs, mineralocorticoid receptor antagonists; LMWH, low molecular weight heparin; VKA, vitamin K antagonists; NOAC, novel oral anticoagulants; SGLT2 inhibitors, sodium glucose co-transporter-2 inhibitors; OMNIBUS, analysis of variance; N/A, non-applicable; ^a^ low risk vs. medium risk, ^b^ low risk vs. high risk, ^c^ medium risk vs. high risk. Red color text = statistically significant values.

**Table 3 viruses-14-00628-t003:** Patient-reported symptoms, vital signs and abnormalities measured during physical examination at hospital admission in the studied cohort.

VariablesUnits	Low Risk [0–1]		Medium [2–3]		High Risk [≥4]		OMNIBUS *p* Value		*p* Value for Post-Hoc Analysis	
	Females N = 682	Males N = 735	Females N = 284	Males N = 208	Females N = 135	Males N = 139	Females	Males	Females	Males
Patient-reported symptoms										
**Cough** *n*/*n*(%)	219 (32.1)	236 (32.1)	71 (25.0)	53 (25.5)	27 (20.0)	42 (30.2)	0.0047	0.1859	0.102 ^a^ 0.0208 ^b^ 0.9427 ^c^	*n*/A
**Dyspnoea** *n*/*n*(%)	244 (35.8)	325 (44.2)	110 (38.7)	96 (46.2)	63 (46.7)	83 (59.7)	0.0551	0.0035	N/A	1.0 ^a^ 0.0033 ^b^ 0.0538 ^c^
**Chest pain** *n*/*n*(%)	49 (7.2)	53 (7.2)	18 (6.3)	16 (7.7)	11 (8.1)	16 (11.5)	0.7855	0.2237	N/A	N/A
**Smell dysfunction** *n*/*n*(%)	26 (3.8)	35 (4.8)	3 (1.1)	7 (3.4)	0 (0.0)	5 (3.6)	0.0039	0.6142	0.0656 ^a^ 0.0414 ^b^ 1.0 ^c^	N/A
**Diarrhoea** *n*/*n*(%)	37 (5.4)	38 (5.2)	22 (7.7)	11 (5.3)	11 (8.1)	8 (5.8)	0.2667	0.9606	N/A	N/A
**Nausea/Vomiting** *n*/*n*(%)	36 (5.3)	21 (2.9)	18 (6.3)	9 (4.3)	11 (8.1)	3 (2.2)	0.4065	0.4662	N/A	N/A
**Measured vital signs**										
**Body temperature**, °C mean ± SD/min-max/N	37.1 ± 0.8 35.0–40.5 416	37.1 ± 0.9 34.4–40.0 393	36.9 ± 0.9 35.8–40.0 131	36.9 ± 1.0 35.0–40.0 104	36.8 ± 0.9 35.2–40.0 63	37.1 ± 0.8 35.5–40.0 78	0.0456	0.3888	0.3 ^a^ 0.07 ^b^ 0.588 ^c^	N/A
**Heart rate**, beats/minute mean ± SD/min-max/N	85.9 ± 14.6 48–150 490	86.9 ± 16.5 48–160 555	84.6 ± 17.2 50–160 217	83.5 ± 15.5 52–140 170	87.4 ± 21.3 36–170 116	82.3 ± 15.8 58–140 124	0.4159	0.0035	N/A	0.045 ^a^ 0.012 ^b^ 0.773 ^c^
**Respiratory rate breaths/minute** mean ± SD/min-max/N	17.9 ± 5.9 12–50 107	18.9 ± 5.7 12–50 97	17.8 ± 3.8 12–31 34	19.6 ± 6.7 12–45 34	19.0 ± 4.1 12–29 22	19.6 ± 7.6 12–50 24	0.5185	0.8014	N/A	N/A
**Systolic blood pressure mmHg** mean ± SD/min-max/N	128.6 ± 21.3 74–240 488	132.6 ± 21.1 60–220 552	133.2 ± 24.2 50–210 216	135.6 ± 26.7 50–270 169	135.6 ± 25.5 70–210 117	133.5 ± 24.0 85–200 127	0.004	0.4149	0.042 ^a^ 0.018 ^b^ 0.687 ^c^	N/A
**Diastolic blood pressure**, mmHg mean ± SD/min-max/N	77.4 ± 12.5 40–150 487	79.5 ± 12.7 40–130 550	77.1 ± 13.7 40–157 214	79.3 ± 13.5 45–150 166	7.5 ± 15.5 40–143 117	75.1 ± 15.2 40–120 127	0.8167	0.0091	N/A	0.986 ^a^ 0.007 ^b^ 0.034 ^c^
**SpO2 on room air**, % (FiO2 = 21%) mean ± SD/min-max/N	94.4 ± 5.9 56–100 421	91.1 ± 7.9 48–99 393	90.8 ± 8.5 50–100 160	88.2 ± 10.9 50–99 121	91.2 ± 6.9 64–99 84	89.2 ± 9.9 50–99 83	<0.0001	0.0102	<0.0001 ^a^ 0.0003 ^b^ 0.934 ^c^	0.018 ^a^ 0.205 ^b^ 0.79 ^c^
**Abnormalities detected during physical examination**										
**Cracles** *n*/*n*(%)	62 (9.1)	92 (12.5)	47 (16.5)	52 (25.0)	30 (22.2)	36 (25.9)	<0.0001	<0.0001	0.0038 ^a^ <0.0001 ^b^ 0.6164 ^c^	<0.0001 ^a^ 0.0002 ^b^ 1.0 ^c^
**Wheezing** *n*/*n*(%)	32 (4.7)	62 (8.4)	23 (8.1)	33 (15.9)	32 (23.7)	37 (26.6)	<0.0001	<0.0001	0.1611 ^a^ <0.0001 ^b,c^	0.0078 ^a^ <0.0001 ^b^ 0.0628 ^c^
**Pulmonarycongestion** *n*/*n*(%)	70 (10.3)	114 (15.5)	51 (18.0)	54 (26.0)	37 (27.4)	41 (29.5)	<0.0001	<0.0001	0.0044 ^a^ <0.0001 ^b^ 0.1096 ^c^	0.0022 ^a^ 0.0004 ^b^ 1.0 ^c^

Categorized variables are presented as: a number with a percentage. Continuous variables are presented as: mean ± SD, range (minimum -maximum) and number of non-missing values. Information about the numbers with valid values is provided in the left column. Abbreviations: N, valid measurements; *n*, number of patients with parameter above the cut-off point; SD, standard deviation. OMNIBUS, analysis of variance; N/A, non-applicable, ^a^ low risk vs. medium risk, ^b^ low risk vs. high risk, ^c^ medium risk vs. high risk. Red color text = statistically significant values.

**Table 4 viruses-14-00628-t004:** Patient initial and on discharge laboratory assay in the studied cohort after C_2_HEST risk stratification.

Parameter Time of Assessment	Units	Low Risk [0–1]		Medium [2–3]		High Risk [≥4]		*p-*Value OMNIBUS		*p-*Value for Post-Hoc Analysis	
		Females	Males	Females	Males	Females	Males	Females	Males	Females	Males
Morphology											
**Leucocytes** *n*/*n*(%)/N **On admission**	>12 × 10^3^/µL	85 (13.8) 615	116 (16.9) 686	52 (18.8) 277	32 (15.8) 203	23 (17.7) 130	29 (212) 137	0.3085	0.3279	N/A	N/A
	4–12× 10^3^/µL	467 (75.9) 615	504 (73.5) 686	198 (71.5) 277	147 (72.4) 203	91 (70.0) 130	100 (73.0) 137				
	<4 × 10^3^/µL	63 (10.2) 615	66 (9.6) 686	27 (9.7) 277	24 (11.8) 203	16 (12.3) 130	8 (5.8) 137				
**On discharge**	>12 × 10^3^/µL	81 (13.2) 615	119 (17.3) 686	55 (19.9) 277	48 (23.6) 203	36 (27.7) 130	28 (20.4) 137	0.0008	0.0028	0.0971 ^a^ 0.0006 ^b^ 0.5375 ^c^	0.002 ^a^ 1.0 ^b^ 0.1331 ^c^
	4–12× 10^3^/µL	487 (79.2) 615	530 (77.3) 686	205 (74.0) 277	132 (65.0) 203	85 (65.4) 130	103 (75.2) 137				
	<4 × 10^3^/µL	47 (7.6) 615	37 (5.4) 686	17 (6.1) 277	23 (11.3) 203	9 (6.9) 130	6 (4.4) 137				
**Haemoglobin** *n*/*n*(%)/N **On admission**	<12 g/dL females <13 g/dL males anaemia	172 (28.0) 615	173 (25.2) 686	91 (32.9) 277	104 (51.2) 203	63 (48.5) 130	84 (61.3) 137	<0.0001	<0.0001	0.4836 ^a^ <0.0001 ^b^ 0.0106 ^c^	<0.0001 ^a,b^ 0.2546 ^c^
**On discharge**		266 (43.3) 615	244 (35.6) 686	122 (44.0) 277	136 (67.0) 203	79 (60.8) 130	92 (67.2) 137	0.0011	<0.0001	1.0 ^a^ 0.0012 ^b^ 0.0071 ^c^	<0.0001 ^a,b^ 1.0 ^c^
**Platelets** mean ± SD/min-max/N **On admission**	×10^3^/µL	244.8 ± 115.7 4.0–1356 615	227.4 ± 101.0 0.0–746.0 686	244.9 ± 115.8 41.0–740.0 277	209.8 ± 108.3 3.0–730.0 203	236.9 ± 98.7 8.0–537.0 130	198.9 ± 83.6 15.0–578.0 137	0.7077	0.001	N/A	0.099 ^a^ 0.002 ^b^ 0.548 ^c^
**On discharge**		267.7 ± 122.9 2.0–929.0 614	273.6 ± 133.0 6.0–1101.0 685	259.6 ± 117.1 27.0–694.0 277	225.7 ± 124.3 3.0–606.0 203	225.6 ± 102.3 4.0–592.0 130	203.3 ± 92.3 15.0–472.0 137	0.0003	<0.0001	0.614 ^a^ 0.0002 ^b^ 0.009 ^c^	<0.0001 ^a,b^ 0.139 ^c^
**Acid -base balance in the arterial blood gas**											
**PH** mean ± SD/min-max/N **On admission**		7.42 ± 0.08 7.19–7.58 48	7.43 ± 0.09 7.04–7.57 73	7.43 ± 0.07 7.24–7.53 37	7.43 ± 0.07 7.10–7.54 51	7.39 ± 0.08 7.09–7.52 32	7.42 ± 0.07 7.28–7.54 35	0.2287	0.8496	N/A	N/A
**On discharge**		7.43 ± 0.07 7.22–7.54 48	7.42 ± 0.09 7.06–7.54 73	7.43 ± 0.06 7.27–7.53 37	7.42 ± 0.09 7.01–7.55 51	7.44 ± 0,06 7.26–7.56 32	7.40 ± 0.06 7.25–7.52 35	0.8782	0.5746	N/A	N/A
**PaO2** mean ± SD/min-max/N **On admission**	mmHg	75.3 ± 33.0 12.8–207.0 48	70.2 ± 22.8 23.5–136.0 73	80.7 ± 54.2 28.3–286.0 37	73.2 ± 42.5 28.6–298.0 51	70.7 ± 25.7 32.8–134.0 32	70.5 ± 41.4 23.7–222.0 35	0.562	0.9031	N/A	N/A
**On discharge**		74.8 ± 27.7 12.8–207..0 48	75.7 ± 26.0 23.5–165.0 73	81.9 ± 55.0 23.3–286.0 37	74.6 ± 43.5 28.6–298.0 51	69.5 ± 27.6 28.5–134.0 32	63.6 ± 20.5 28.5–129.0 35	0.4499	0.0316	N/A	0.985 ^a^ 0.028 ^b^ 0.268 ^c^
**PaCO2** mean ± SD/min-max/N **On admission**	mmHg	38.3 ± 8.2 20.2–58.0 48	37.8 ± 11.5 25.7–82.4 73	37.2 ± 9.3 26.9–79.4 37	36.3 ± 9.6 20.9–67.0 51	38.6 ± 13.6 25.0–88,4 32	38.7 ± 8.0 19.7–61.0 35	0.8084	0.4415	N/A	N/A
**On discharge**		38.3 ± 8.4 20.2–62.2 48	38.5 ± 10.7 24.1–75.5 73	38.5 ± 10.0 27.8–84.4 37	37.5 ± 11.7 20.9–88.4 51	37.4 ± 11.5 25.0–88.4 32	39.9 ± 8.7 26.8–67.8 35	0.9071	0.5398	N/A	N/A
**HCO3 standard** mean ± SD/min-max/N **On admission**	mmol/L	25.0 ± 3.7 12.5–32.9 47	24.9 ± 3.8 12.1–32.8 73	24.9 ± 4.4 16.9–39.5 36	24.0 ± 4.0 14.3–32.4 49	23.4 ± 4.6 13.5–32.3 32	24.8 ± 4.5 17.5–38.6 35	0.2666	0.4967	N/A	N/A
**On discharge**		25.3 ± 3.4 12.5–35.7 47	24.8 ± 4.0 12.1–33.6 73	25.7 ± 4.8 16.9–40.3 36	25.0 ± 6.1 13.7–51.7 49	25.1 ± 4.3 17.4–35.8 32	24.7 ± 3.7 19.4–36.7 35	0.8862	0.9539	N/A	N/A
**BE** mean ± SD/min-max/N **On admission**	mmol/L	0.63 ± 5.06 [−]15.7–5.9 16	1.12 ± 4.67 [−]9.1–10.5 25	2.96 ± 4.72 [−]3.3–15.7 17	0.88 ± 5.59 [−]12.5–9.7 26	[−]0.1 ± 4.75 [−]7.4–7.9 7	2.92 ± 5.21 [−]3.3–14.6 17	0.2745	0.4315	N/A	N/A
**On discharge**		1.21 ± 5.91 [−]15.7–11.9 16	0.46 ± 5.21 [−]11.0–8.3 25	3.54 ± 4.99 [−]3.3–17.1 17	1.62 ± 6.58 [−]14.7–11.8 26	0.91 ± 4.58 [−]7.4–7.9 7	1.65 ± 5.0 [−]5.3–13.2 17	0.363	0.6978	N/A	N/A
**Lactates** mean ± SD/min-max/N **On admission**	mmol/L	2.0 ± 0.8 0.6–4.3 38	2.7 ± 1.9 1.1–12.8 67	2.0 ± 1.0 0.6–5.7 32	2.0 ± 0.7 0.5–3.8 47	2.9 ± 2.1 0.8–12.0 31	2.1 ± 1.4 0.6–5.7 30	0.1027	0.0291	N/A	0.02 ^a^ 0.199 ^b^ 0.913 ^c^
**On discharge**		2.1 ± 0.8 0.7–4.9 38	2.7 ± 1.9 1.0–12.8 67	2.0 ± 0.9 0.6–5.7 32	2.2 ± 1.1 0.5–6.4 47	2.6 ± 1.3 0.8–6.0 31	2.2 ± 1.1 0.8–4.3 30	0.0544	0.239	N/A	N/A
**Electrolytes, inflammatory and iron biomarkers**											
**Na** mean ± SD/min-max/N **On admission**	mmol/L	138.3 ± 3.8 106.0−155.0 605	138.2 ± 4.8 109.0−159.0 683	137.7 ± 7.6 101.0−175.0 272	137.7 ± 6.1 105.0−158.0 203	138.3 ± 7.7 108.0−174.0 130	137.6 ± 5.9 112.0−158.0 137	0.4803	0.3745	N/A	N/A
**On discharge**		138.9 ± 3.7 113.0−167.0 605	139.3 ± 4.8 109.0−175.0 683	139.0 ± 7.4 101.0−172.0 272	139.4 ± 7.2 105.0−165.0 203	140.7 ± 7.1 124.0−172.0 130	139.8 ± 6.3 120.0–157.0 137	0.0179	0.6389	0.977 ^a^ 0.013 ^b^ 0.062 ^c^	N/A
**K** mean ± SD/min-max/N **On admission**	mmol/L	3.99 ± 0.54 2.33–6.5 609	4.13 ± 0.61 2.0–7.5 684	4.06 ± 0.7 2.42 ± 5.9 275	4.25 ± 0.69 2.4–7.0 202	4.14 ± 0.74 2.53–6.6 130	4.43 ± 0.87 3.0–8.7 137	0.0403	0.0002	0.325 ^a^ 0.059 ^b^ 0.479 ^c^	0.072 ^a^ 0.0005 ^b^ 0.1 ^c^
**On discharge**		4.13 ± 0.56 2.47–7.4 609	4.33 ± 0.6 2.0–6.9 684	4.26 ± 0.75 2.28–6.32 275	4.5 ± 0.77 2.4–7.0 202	4.36 ± 0.69 2.53–6.5 130	4.51 ± 0.69 2.76–6.64 137	0.0004	0.0011	0.033 ^a^ 0.002 ^b^ 0.373 ^c^	0.015 ^a,b^ 0.983 ^c^
**CRP** mean ± SD/min-max/N **On admission**	mg/L	60.49 ± 72.41 0.13−531.58 597	90.54 ± 91.63 0.32−496.98 677	74.25 ± 84.61 0.4−538.55 275	95.36 ± 88.06 0.29–487.38 202	64.75 ± 72.93 0.4–344.95 130	87.45 ± 87.37 0.4–390.94 137	0.0674	0.69258	N/A	N/A
**On discharge**		36.85 ± 64.5 0.13–494.73 597	58.33± 88.96 0.25–496.98 677	62.6 ± 89.56 0.22–538.55 275	86.23± 99.39 0.46–447.61 202	63.78± 80.7 0.4–431.9 130	83.42± 90.91 0.42–390.94 137	<0.0001	0.0001	<0.0001 ^a^ 0.001 ^b^ 0.99 ^c^	0.001 ^a^ 0.01 ^b^ 0.961 ^c^
**Procalcitonin** mean ± SD/min-max/N **On admission**	ng/mL	0.33 ± 1.55 0.01–24.95 404	1.24 ± 5.79 0.01–61.28 514	2.0 ± 15.13 0.01–196.04 188	1.62 ± 6.6 0.01–72.61 156	1.36 ± 6.46 0.01–60.77 98	1.59 ± 5.81 0.01–49.83 113	0.0993	0.7214	N/A	N/A
**On discharge**		0.57 ± 3.26 0.01–41.32 404	1.16 ± 6.14 0.01–75.16 514	0.86 ± 3.62 0.01–30.67 188	2.49 ± 8.44 0.01–81.09 156	1.11 ± 6.17 0.01–60.77 98	1.19 ± 3.68 0.01–27.61 113	0.5044	0.1807	N/A	N/A
**IL-6** mean ± SD/min-max/N **On admission**	pg/mL	85.5 ± 660.2 2.0–9099.0 192	45.2 ± 98.7 2.0–1000.0 288	34.3 ± 52.7 2.0–398.0 84	55.9 ± 75.3 2.0–499.0 59	55.2 ± 94.1 2.0–421.0 38	69.2 ± 97.8 2.0–369.0 40	0.2692	0.2811	N/A	N/A
**On discharge**		90.3 ± 672.0 2.0–9099.0 192	42.0 ± 111.0 2.0–1000.0 288	28.5 ± 53.5 2.0–398.0 84	56.5 ± 94.3 2.0–499.0 59	67.6 ± 170.4 2.0–1000.0 38	82.3 ± 150.6 2.0–804.0 40	0.1877	0.1939	N/A	N/A
**D-dimer** mean ± SD/min-max/N **On admission**	µg/mL	2.60 ± 8.39 0.15–-118.32 444	4.63 ± 14.46 0.18–132.82 558	5.40 ± 12.57 0.2–107.65 206	7.84 ± 20.75 0.23–127.24 167	3.78 ± 11.48 0.24–107.54 100	7.01 ± 21.41 0.22–128.0 103	0.0133	0.1192	0.011 ^a^ 0.596 ^b^ 0.501 ^c^	N/A
**On discharge**		3.17 ± 11.99 0.15–128.0 444	3.25 ± 9.63 0.21–115.13 558	4.38 ± 8.28 0.21–74.28 206	7.2 ± 17.51 0.23–106.02 167	3.65 ± 11.23 0.21–107.54 100	3.72 ± 6.9 0.22–46.72 103	0.3287	0.0215	N/A	0.016 ^a^ 0.821 ^b^ 0.059 ^c^
**INR** mean ± SD/min-max/N **On admission**		1.07 ± 0.2 0.82–3.6 580	1.19 ± 0.63 0.83–15.2 647	1.25 ± 0.69 0.87–7.8 257	1.27 ± 0.44 0.89–4.37 188	1.58 ± 1.75 0.9–18.74 127	1.99 ± 2.98 0.89–21.1 124	<0.0001	0.0031	0.0002 ^a^ 0.005 ^b^ 0.112 ^c^	0.136 ^a^ 0.01 ^b^ 0.023 ^c^
**On discharge**		1.1 ± 0.4 0.82–9.2 580	1.17 ± 0.33 0.87–6.82 647	1.2 ± 0.8 0.88–13.1 257	1.32 ± 0.7 0.92–7.85 188	1.4 ± 0.8 0.9–8.0 127	1.53 ± 1.88 0.87–21.1 124	0.0003	0.0019	0.048 ^a^ 0.001 ^b^ 0.251 ^c^	0.011 ^a^ 0.082 ^b^ 0.452 ^c^
**APTT** *n*/*n*(%)/N **On admission**	>60 s	6 1.1 561	22 3.5 630	3 1.2 247	4 2.2 184	6 4.8 124	5 4.2 120	0.0243	0.5704	1.0 ^a^ 0.0337 ^b^ 0.1964 ^c^	N/A
**On discharge**		14 2.5 561	32 5.1 630	3 1.2 247	5 2.7 184	4 3.2 124	8 6.7 120	0.3472	0.2518	N/A	N/A
**Fibrinogen** mean ± SD/min-max/N **On admission**	g/dL	4.69 ± 1.53 0.35–9.04 153	5.11 ± 2.14 0.44–10.0 132	4.34 ± 1.4 0.35–6.72 29	4.93 ± 2.0 0.37–9.2 52	3.62 ± 1.06 1.78–5.51 24	5.31 ± 1.71 2.54–9.1 29	0.0004	0.6765	0.441 ^a^ 0.0003 ^b^ 0.096 ^c^	N/A
**On discharge**		4.58 ± 1.8 0.44–10.0 153	4.95 ± 2.13 0.6–10.0 132	5.01 ± 2.11 0.35–9.4 29	4.98 ± 2.3 0.37–11.3 52	3.84 ± 1.21 1.53–5.75 24	5.71 ± 2.07 2.2–9.04 29	0.0184	0.2055	0.561 ^a^ 0.037 ^b^ 0.04 ^c^	N/A

Continuous variables are presented as: mean ± SD. range (minimum -maximum) and number of non-missing values. Categorized variables are presented as: a number with a percentage. Information about the numbers with valid values is provided in the left column. Abbreviations: N, valid measurements; n, number of patients with parameter above cut-off point; SD, standard deviation. OMNIBUS, analysis of variance; N/A, non-applicable, ^a^ low risk vs. medium risk, ^b^ low risk vs. high risk, ^c^ medium risk vs. high risk. Red text—statistically significant values.

**Table 5 viruses-14-00628-t005:** Patient initial and on discharge laboratory assay in the studied cohort after C_2_HEST risk stratification.

Parameter Time of Assessment	Units	Low Risk [0–1]		Medium [2–3]		High Risk [≥4]		*p-*Value OMNIBUS		*p-*Value for Post-Hoc Analysis	
		Females	Males	Females	Males	Females	Males	Females	Males	Females	Males
Biochemistry											
**Glucose** mean ± SD/min-max/N **On admission**	mg/dL	128.1 ± 67.0 61.0–671.0 425	139.3 ± 79.5 28.0–933.0 638	144.1 ± 74.9 54.0–662.0 257	160.5 ± 110.3 47.0–1026.0 192	149.1 ± 86.5 70–685 120	152.0 ± 109.4 37.0–1064.0 126	0.0035	0.0315	0.014 ^a^ 0.039 ^b^ 0.849 ^c^	0.038 ^a^ 0.433 ^b^ 0.779 ^c^
**On discharge**		119.0 ± 56.0 37.0–595.0 425	127.3 ± 78.8 50.0–1444.0 638	136.4 ± 75.3 54.0–596.0 257	150.7 ± 92.2 47.0–578.0 192	144.8 ± 90.4 14.0–685.0 120	143.5 ± 63.1 37.0–406.0 126	0.0003	0.0012	0.004 ^a^ 0.01 ^b^ 0.653 ^c^	0.005 ^a^ 0.033 ^b^ 0.688 ^c^
**Glycated hemoglobin (HbA1c**) mean ± SD/min-max/N **On admission**	%	7.1 ± 1.9 4.2–12.2 47	7.9 ± 2.5 4.9–14.9 80	7.9 ± 2.7 4.9–16.6 39	7.2 ± 1.4 4.8–12.2 36	7.2 ± 1.7 5.1–11.4 33	7.4 ± 1.9 5.1–13.7 28	0.3182	0.1497	N/A	N/A
**On discharge**		7.0 ± 1.8 4.2–12.2 47	7.8 ± 2,4 4.9–14.9 80	7.9 ± 2.7 4.9–16.8 39	7.1 ± 1.4 4.7–12.2 36	7.2 ± 1.7 5.1–11.4 33	7.4 ± 1.9 5.1–13.7 28	0.2299	0.1563	N/A	N/A
**Urea** mean ± SD/min-max/N **On admission**	mg/dL	36.3 ± 35.1 7.0–301.0 481	47.6 ± 35.8 5.0–307.0 664	60.2 ± 50.6 8.0–353.0 256	69.9 ± 47.5 15.0–271.0 199	69.5 ± 48.9 12.0–336.0 124	84.4 ± 57.1 17.0–369.0 133	<0.0001	<0.0001	<0.0001^a,b^ 0.197 ^c^	<0.0001^a,b^ 0.042 ^c^
**On discharge**		35.5 ± 29.6 7.0–231.0 481	44.9 ± 32.9 5.0–307.0 664	59.0 ± 48.2 10.0–353.0 256	75.6 ± 59.8 12.0–396.0 199	66.9 ± 41.7 15.0–204.0 124	88.9 ± 58.6 21.0–342.0 133	<0.0001	<0.0001	<0.0001^a,b^ 0.236 ^c^	<0.0001^a,b^ 0.11 ^c^
**Creatinine** mean ± SD/min-max/N **On admission**	mg/dL	1.0 ± 0.99 0.34–11.99 533	1.26 ± 1.3 0.26–14.87 683	1.22 ± 0.97 0.48–9.56 275	1.76 ± 1.6 0.58–12.66 203	1.58 ± 1.27 0.44–8.46 130	2.02 ± 1.81 0.49–11.3 137	<0.0001	<0.0001	0.008 ^a^ < 0.0001 ^b^ 0.012 ^c^	0.0002 ^a^ < 0.0001 ^b^ 0.369 ^c^
**On discharge**		0.96 ± 0.86 0.34–9.11 533	1.16 ± 1.18 0.26–14.87 683	1.16 ± 0.92 0.45–9.06 275	1.81 ± 1.72 0.43–12.35 203	1.42 ± 1.21 0.43–7.66 130	1.89 ± 1.58 0.43–9.27 137	<0.0001	<0.0001	0.009 ^a^ 0.0002 ^b^ 0.084 ^c^	<0.0001^a,b^ 0.877 ^c^
**eGFR** mean ± SD/min-max/N **On admission**	mL/min/1.73 m^2^	84.6 ± 32.1 0.0–207.0 531	85.3 ± 35.9 3.0–433.0 680	60.8 ± 25.0 4.0–136.0 275	63.7 ± 33.1 4.0–149.0 203	49.7 ± 26.4 5.0–145.0 130	55.3 ± 32.0 5.0–180.0 137	<0.0001	<0.0001	0.0 ^a^ < 0.0001 ^b^ 0.0002 ^c^	0.0 ^a^ 0.0 ^b^ 0.054 ^c^
**On discharge**		86.6 ± 32.1 0.0–207.0 531	91.5 ± 36.5 3.0–433.0 680	65.0 ± 26.6 4.0–148.0 275	66.0 ± 36.1 4.0–208,0 203	58.2 ± 30.3 5.0–147.0 130	58.6 ± 35.7 6.0–209.0 137	<0.0001	<0.0001	0.0 ^a^ < 0.0001 ^b^ 0.076 ^c^	<0.0001^a,b^ 0.147 ^c^
**Total protein** mean ± SD/min-max/N **On admission**	g/L	6.1 ± 0.8 3.9–8.2 145	6.1 ± 0.8 3.5–8.1 186	5.8 ± 0.8 3.6–8.2 78	6.0 ± 1.0 4.2–9.5 74	5.7 ± 0.9 3.3–8.1 62	5.7 ± 0.9 3.4–8.2 61	0.0235	0.0555	0.148 ^a^ 0.033 ^b^ 0.741 ^c^	N/A
**On discharge**		6.0 ± 0.9 3.9–8.2 145	6.0 ± 0.9 3.0–8.1 186	5.7 ± 0.9 3.7–8.2 78	5.9 ± 0.9 4.3–9.1 74	5.5 ± 1.0 3.3–8.1 62	5.7 ± 0.9 3.4–7.8 61	0.0012	0.0162	0.049 ^a^ 0.002 ^b^ 0.388 ^c^	0.799 ^a^ 0.012 ^b^ 0.158 ^c^
**Albumin** mean ± SD/min-max/N **On admission**	g/L	3.1 ± 0.6 1.6–4.6 152	3.2 ± 0.6 1.5–5.1 222	3.0 ± 0.5 1.1–4.3 78	3.2 ± 0.6 2.1–4.4 82	2.9 ± 0.6 0.7–3.7 62	3.1 ± 0.6 1.5–4.9 67	0.0134	0.3087	0.287 ^a^ 0.011 ^b^ 0.307 ^c^	N/A
**On discharge**		3.1 ± 0.6 1.1–4.6 152	3.0 ± 0.7 0.4–5.1 222	3.0 ± 0.5 1.9–4.2 78	3.1 ± 0.6 1.7–4,4 82	2.8 ± 0.5 1.4–3.7 62	2.8 ± 0.7 0.9–4..5 67	0.005	0.0549	0.64 ^a^ 0.004 ^b^ 0.277 ^c^	N/A
**AST** mean ± SD/min-max/N **On admission =**	IU/L	56.8 ± 139.7 6.0–2405.0 384	62.7 ± 89.4 5.0–1261.0 499	72.7 ± 343.6 8.0–4776 193	58.8 ± 49.5 7.0–323.0 154	113.5 ± 450.8 8.0–3866.0 104	60.2 ± 101.8 10.0–731.0 107	0.3869	0.7844	N/A	N/A
**On discharge**		123.4 ± 1244.4 10.0–23,896.0 384	68.3 ± 255.1 5.0–3761.0 499	43.3 ± 46.5 8.0–380.0 193	107.5 ± 537.6 11.0–6591.0 154	148.9 ± 702.4 8.0–6088.0 104	97.4 ± 402.4 7.0–4019.0 107	0.1438	0.5525	N/A	N/A
**ALT** mean ± SD/min-max/N **On admission**	IU/L	47.0 ± 87.7 5.0–1411.0 435	61.4 ± 96.4 4.0–1278.0 537	52.2 ± 251.2 5.0–3700.0 219	45.0 ± 43.2 4.0–270.0 172	57.1 ± 183.6 5.0–1361.0 112	46.7 ± 88.2 6.0–612.0 113	0.8212	0.0081	N/A	0.006 ^a^ 0.256 ^b^ 0.98 ^c^
**On discharge**		65.5 ± 265.4 6.0–5163.0 435	74.3 ± 105.0 4.0–1217.0 537	38.5 ± 46.1 5.0–449.0 219	65.1 ± 124.7 7.0–1247.0 172	74.4 ± 308.8 5.0–2985.0 112	71.4 ± 207.3 9.0–1570.0 113	0.0624	0.6835	N/A	N/A
**Bilirubin** mean ± SD/min-max/N **On admission**	mg/dL	0.78 ± 1.68 0.1–19.1 363	0.88 ± 1.24 0.1–15.1 489	0.85 ± 0.88 0.2–9.2 195	0.80 ± 0.49 0.2–3.1 157	0.77 ± 0.51 0.1–4.2 100	0.98 ± 0.84 0.3–6.6 103	0.5771	0.1292	N/A	N/A
**On discharge**		0.77 ± 1.65 0.1–19.0 363	0.95 ± 1.91 0.1–25.9 489	0.95 ± 2.55 0.2–35.3 195	0.76 ± 0.47 0.2–3.1 157	0.78 ± 0.67 0.3–6.1 100	1.06 ± 1.33 0.2–12.8 103	0.6611	0.0224	N/A	0.123 ^a^ 0.754 ^b^ 0.08 ^c^
**LDH** mean ± SD/min-max/N **On admission**	U/L	404.5 ± 478.5 50.0–7100.0 328	448.6 ± 282.2 120.0–3194.0 448	368.2 ± 189.8 44.0–1357.0 156	418.9 ± 212.9 134.0–1172.0 130	468.1 ± 1015.3 71.0–9505.0 83	416.9 ± 269.7 113.0–1863.0 86	0.3576	0.3427	N/A	N/A
**On discharge**		387.2 ± 739.3 50.0–11,227.0 328	389.2 ± 396.2 93.0–6577.0 448	340.3 ± 167.3 44.0–1357.0 156	407.1 ± 243.5 112.0–1584.0 130	474.0 ± 1028.1 106.0–9505.0 83	388.8 ± 215.4 97.0–1260.0 86	0.292	0.7848	N/A	N/A
**Cardiacbiomarkers**											
**BNP** mean ± SD/min-max/N **On admission**	pg/mL	152.5 ± 241.1 1.7–1130.8 54	254.1 ± 763.7 1.7–6924.2 107	455.4 ± 872.4 10.1–4890.6 50	433.3 ± 747.2 3.0–3153.2 50	711.7 ± 995.6 22.3–4993.0 56	1432.8 ± 2864.5 5.9–13,368.4 42	<0.0001	0.0206	0.054 ^a^ 0.0004 ^b^ 0.338 ^c^	0.35 ^a^ 0.031 ^b^ 0.082 ^c^
**On discharge**		177.7 ± 308.1 5.3–1877.0 54	239.8 ± 753.1 1.7–6924.2 107	536.1 ± 1562.6 10.1–10,622.8 50	396.2 ± 697.6 3.0–3153.2 50	592.3 ± 769.1 22.3–3729.8 56	1389.2 ± 2735.4 11.9–13,368.4 42	0.0008	0.0206	0.257 ^a^ 0.001 ^b^ 0.971 ^c^	0.412 ^a^ 0.027 ^b^ 0.067 ^c^
**NT-proBNP** mean ± SD/min-max/N **On admission**	ng/mL	1467.1± 3250.7 18.7–16,551.7 62	2126.5± 9426.7 12.0–70,000.0 110	6608.9± 12,708.7 49.6–70,000.0 54	10,323.4 ± 16,141.4 18.2–70,000.0 55	14,888.1 ± 18,982.5 119.6–70,000.0 43	13,522.6 ± 19,276.7 343.7–70,000.0 55	<0.0001	<0.0001	0.015 ^a^ 0.0001 ^b^ 0.043 ^c^	0.002 ^a^ 0.0003 ^b^ 0.614 ^c^
**On discharge**		1694.0 ± 5047.8 28.5–35,000.0 62	1893.4 ± 7660.6 12.0–70,000.0 110	7852.3 ± 15,159.0 49.6–70,000.0 54	10,661.5 ± 16,202.2 18.2–70,000.0 55	13,084.8 ± 17,275.9 119.6–69,519.7 43	13,265.6 ± 17,873.3 391.3–70,000.0 55	<0.0001	<0.0001	0.016 ^a^ 0.0003 ^b^ 0.267 ^c^	0.0009 ^a^ <0.0001 ^b^ 0.703 ^c^
**Troponin I**, mean ± SD/min-max/N **On admission**	ng/mL	53.1 ± 211.1 0.0–1994.8 263	189.6 ± 1015.9 1.3–11,758.2 415	658.5 ± 7215.3 1.9–94,365.5 171	3044.2 ± 15,485.9 1.0–125,592.6 134	988.4 ± 3316.8 3.3–21,022.9 94	542.0 ± 1724.6 4.8–14,128.8 97	0.015	0.0185	0.517 ^a^ 0.02 ^b^ 0.867 ^c^	0.087 ^a^ 0.133 ^b^ 0.156 ^c^
**On discharge**		105.7 ± 873.1 0.2–12,391.6 263	124.0 ± 797.8 0.8–11,758.2 415	692.7 ± 7243.6 1.9–94,365.5 171	3359.3 ± 18,244.2 0.8–174,652.6 134	838.2 ± 3666.2 1.8–29.828.3 94	493.1 ± 1504.8 4.8–12,657.2 97	0.0977	0.0095	N/A	0.104 ^a^ 0.055 ^b^ 0.17 ^c^
*n*/*n*(%)/N = F: >46.8 ng/mL M: >102.6 ng/mL	>3-fold upper range	46 17.5 263	67 16.1 415	51 29.8 171	47 35.1 134	49 52.1 94	38 39.2 97	<0.0001	<0.0001	0.0113 ^a^ <0.0001^b^ 0.0017 ^c^	<0.0001^a,b^ 1.0 ^c^
**LDL-cholesterol** mean ± SD/min-max/N **On admission**	mg/dL	106.8 ± 64.8 6.0–510.0 85	96.2 ± 40.5 27.0–242.0 147	93.9 ± 39.7 23.0–199.0 69	79.4 ± 40.6 17.0–230.0 60	83.3 ± 44.2 14.0–187.0 49	64.2 ± 37.6 6.0–210.0 39	0.0498	<0.0001	0.283 ^a^ 0.038 ^b^ 0.381 ^c^	0.022 ^a^ <0.0001 ^b^ 0.142 ^c^
**HDL-cholesterol** mean ± SD/min-max/N **On admission**	mg/dL	43.9 ± 17.9 2.0–120.0 86	37.7 ± 14.5 10.0–101.0 150	44.5 ± 16.7 12.0–110.0 69	35.2 ± 11.9 7.0–66.0 60	39.8 ± 17.5 8.0–79.0 48	34.0 ± 10.3 17.0–61.0 38	0.303979	0.154387	N/A	N/A
**Triglycerides** mean ± SD/min-max/N **On admission**	mg/dL	189.4 ± 154.5 40.0–1100.0 122	173.7 ± 105.1 44.0–664.0 237	141.0 ± 94.5 48.0–595.0 83	148.0 ± 98.8 50.0–550.0 81	133.4 ± 56.7 46.0–282.0 60	124.8 ± 66.9 51.0–413.0 56	0.0022	0.0001	0.016 ^a^ 0.001 ^b^ 0.817 ^c^	0.117 ^a^ <0.0001 ^b^ 0.232 ^c^
**Hormones**											
**25-hydroxy-vitamin D** mean ± SD/min-max/N **On admission**	ng/mL	27.4 ± 21.8 3.5–146.1 99	23.4 ± 15.0 3.5–126.4 206	26.1 ± 17.2 3.5–77.7 63	22.9 ± 15.4 5.1–75.6 45	22.4 ± 16.8 3.5–63.5 36	14.5 ± 9.6 3.5–39.1 25	0.3738	0.0006	N/A	0.974 ^a^ 0.0006 ^b^ 0.018 ^c^
**TSH** mean ± SD/min-max/N **On admission**	mIU/L	1.55 ± 2.0 0.01–18.6 186	1.2 ± 1.06 0.0–6.33 255	1.72 ± 2.98 0.01–28.81 137	1.31 ± 1.39 0.01–8.28 95	2.74 ±5.04 0.0–38.24 85	1.43 ± 1.25 0.0–6.36 62	0.1063	0.3834	N/A	N/A

Continuous variables are presented as: mean ± SD. range (minimum -maximum) and number of non-missing values. Categorized variables are presented as: a number with a percentage. Information about the numbers with valid values is provided in the left column. Abbreviations: N, valid measurements; n, number of patients with parameter above cut-off point; SD, standard deviation. OMNIBUS, analysis of variance; N/A, non-applicable, ^a^ low risk vs. medium risk, ^b^ low risk vs. high risk, ^c^ medium risk vs. high risk. Red text = statistically significant values.

**Table 6 viruses-14-00628-t006:** Treatment applied during hospitalization.

Variables, Units	Low Risk [0–1]		Medium [2–3]		High Risk [≥4]		*p-*Value OMNIBUS		*p* Value for Post-Hoc Analysis	
	Females N = 682	Males N = 735	Females N = 384	Males N = 208	Females N = 135	Males N = 139	Females	Males	Females	Males
Applied treatment and procedures										
**Systemic corticosteroid** *n*/*n*(%)	299 (43.8)	409 (55.6)	127 (44.7)	119 (57.2)	64 (47.4)	78 (56.1)	0.7456	0.9222	N/A	N/A
**Convalescentplasma** *n*/*n*(%)	54 (7.9)	113 (15.4)	12 (4.2)	29 (13.9)	15 (11.1)	16 (11.5)	0.0274	0.4749	0.1599 ^a^ 0.8816 ^b^ 0.0406 ^c^	N/A
**Tocilizumab** *n*/*n*(%)	11 (1.6)	11 (1.5)	0 (0.0)	2 (1.0)	1 (0.7)	0 (0.0)	0.054	0.4308	N/A	N/A
**Remdesivir** *n*/*n*(%)	83 (12.2)	153 (20.8)	37 (13.0)	35 (16.8)	12 (8.9)	23 (16.5)	0.4627	0.2822	N/A	N/A
**Antibiotic** *n*/*n*(%)	338 (49.6)	408 (55.5)	157 (55.3)	146 (70.2)	88 (65.2)	103 (74.1)	0.0026	<0.0001	0.3633 ^a^ 0.0038 ^b^ 0.2079 ^c^	0.0006 ^a^ 0.0002 ^b^ 1.0 ^c^

Continuous variables are presented as: mean ± SD, range (minimum–maximum) and number of non-missing values. Categorized variables are presented as: a number with a percentage. Information about the numbers with valid values is provided in the left column. Abbreviations: N, valid measurements; *n*, number of patients with parameter above cut-off point; SD, standard deviation; OMNIBUS, analysis of variance; N/A, non-applicable; ^a^ low risk vs. medium risk, ^b^ low risk vs. high risk, ^c^ medium risk vs. high risk. Red text = statistically significant values.

**Table 7 viruses-14-00628-t007:** Applied treatment and procedures.

Variables	Low Risk [0–1]		Medium [2–3]		High Risk [≥4]		*p* Value OMNIBUS		*p* Value for Post-Hoc Analysis	
	Females N = 681	Males N = 734	Females N= 284	Males N = 207	Females N = 135	Males N = 139	Females	Males	Females	Males
Applied treatment and procedures										
**The most advanced respiratory support applied during the hospitalisation** **no oxygen** *n*/*n*(%)	409 (60.1)	332 (45.2)	140 (49.3)	62 (30.0)	50 (37.0)	39 (28.1)	<0.0001	<0.0001	0.001^a^ <0.0001 ^b^ 0.0114 ^c^	0.0001 ^a^ 0.0007^b^ 1.0 ^c^
**low flow oxygen support** *n*/*n*(%)	199 (29.2)	252 (34.3)	103 (36.3)	85 (41.1)	65 (48.1)	59 (42.4)				
**high flow nasal cannula** non-invasive ventilation *n*/*n*(%)	26 (3.8)	56 (7.6)	24 (8.5)	28 (13.5)	17 (12.6)	22 (15.8)				
**invasive ventilation** *n*/*n*(%)	47 (6.9)	94 (12.8)	17 (6.0)	32 (15.5)	3 (2.2)	19 (13.7)				
**Oxygenation parameters from the period of qualification for advanced respiratory support**: **SpO2**, % mean ± SD/(min-max/N	92.2 ± 6.8 (59–100) 221	88.8 ± 8.6 (50–100) 189	87.0 ± 11.0 (55–99) 64	86.0 ± 8.4 (60–99) 69	86.2 ± 9.3 (59–98) 40	85.1 ± 10.5 (60–99) 48	<0.0001	0.0159	0.002 ^a^ 0.0008 ^b^ 0.908 ^c^	0.057 ^a^ 0.072 ^b^ 0.87 ^c^
**Therapy with catecholamines** *n*/*n*(%)/N	39 (5.7) 682	92 (12.5) 735	14 (4.9)	31 (14.9) 208	9 (6.7)	33 (23.7)	0.7614	0.0025	N/A	1.0 ^a^ 0.0026 ^b^ 0.1576 ^c^
**Coronary revascularisation or/and an indication for coronary revascularisation,** *n*/*n*(%)/N	1 (0.1) 682	7 (1.0) 735	3 (1.1)	8 (3.8) 208	1 (0.7)	6 (4.3)	0.0795	0.0021	N/A	0.0225 ^a^ 0.0286 ^b^ 1.0 ^c^
**Haemodialysis** *n*/*n*(%)/N	15 (2.2) 682	31 (4.2) 735	2 (0.7)	11 (0.7) 208	4 (3.0)	8 (5.8)	0.1486	0.6417	N/A	N/A

Continuous variables are presented as: mean ± SD, range (minimum–maximum) and number of non-missing values. Categorized variables are presented as: a number with a percentage. Information about the numbers with valid values is provided in the left column. Abbreviations: N, valid measurements; *n*, number of patients with parameter above cut-off point; SD, standard deviation; OMNIBUS, analysis of variance;N/A, non-applicable; ^a^ low risk vs. medium risk, ^b^ low risk vs. high risk, ^c^ medium risk vs. high risk. Red text = statistically significant values.

**Table 8 viruses-14-00628-t008:** Total and in-hospital all-cause mortality in the C_2_HEST risk strata in males’ and females’ cohort.

Variables	Low Risk [0–1]		Medium [2–3]		High Risk [≥4]		*p* Value OMNIBUS		*p* Value for Post-Hoc Analysis	
	Females N = 682	Males N = 735	Females N = 284	Males N = 208	Females N = 135	Males N = 139	Females	Males	Females	Males
All-cause mortality rate										
**In-hospital mortality** *n*/*n*(%)	36 (5.3)	83 (11.3)	50 (17.6)	60 (28.8)	43 (31.9)	54 (38.8)	<0.0001	<0.0001	<0.0001 ^a,b^ 0.0048 ^c^	<0.0001 ^a,b^ 0.2029 ^c^
**3-month mortality** *n*/*n*(%)	68 (10.0)	134 (18.2)	95 (33.5)	103 (49.5)	65 (48.1)	82 (59.0)	<0.0001	<0.0001	<0.0001 ^a,b^ 0.016 ^c^	<0.0001 ^a,b^ 0.3134 ^c^
**6-month mortality** *n*/*n*(%/)/N	72 (17.3) 415	142 (31.4) 452	104 (49.3) 211	104 (60.1) 173	70 (61.4) 114	86 (68.8) 125	<0.0001	<0.0001	<0.0001 ^a,b^ 0.1454 ^c^	<0.0001 ^a,b^ 0.4696 ^c^
**Hospitalization**										
**Duration of hospitalization days** mean ± SD/(min-max)	10.4 ±12.7 (1–131)	12.4 ± 14.4 (1–130)	12.1 ± 11.9 (1–68)	14.6 ± 15.6 (1–124)	18.3 ±17.5 (1–87)	13.9 ± 13.9 (1–121)	<0.0001	0.1386	0.128 ^a^ <0.0001 ^b^ 0.0007 ^c^	NA
**End of hospitalisation death** *n*/*n*(%)	36 (5.3)	83 (11.3)	50 (17.6)	60 (28.8)	43 (31.9)	54 (38.8)	<0.0001	<0.0001	<0.0001 ^a,b^ 0.0143 ^c^	<0.0001 ^a,b^ 0.3663 ^c^
**discharge to home–full recovery**	515 (75.5)	478 (65.0)	141 (49.6)	79 (38.0)	57 (42.2)	46 (33.1)				
**transfer to another hospital** **–worsening)**	60 (8.8)	79 (10.7)	59 (20.8)	38 (18.3)	17 (12.6)	27 (19.4)				
**transfer to another hospital** **–in recovery**	71 (10.4)	95 (12.9)	34 (12.0)	31 (14.9)	18 (13.3)	12 (8.6)				

Continuous variables are presented as: mean ± SD, range (minimum–maximum) and number of non-missing values. Categorized variables are presented as: a number with a percentage. Information about the numbers with valid values is provided in the left column. Abbreviations: N, valid measurements; *n*, number of patients with parameter above cut-off point; SD, standard deviation; OMNIBUS, analysis of variance; N/A, non-applicable; ^a^ low risk vs. medium risk, ^b^ low risk vs. high risk, ^c^ medium risk vs. high risk. Red text = statistically significant values.

**Table 9 viruses-14-00628-t009:** The total all-cause-death hazard Ratios for C_2_HEST risk stratification in female cohort.

**Total Death**			
	**HR**	**95%CI**	***p* Value**
**Overall**	1.428	1.349–1.513	<0.0001
**Risk strata**			
**Low risk vs. Medium risk**	4.267	3.170–5.732	<0.0001
**Low risk vs. High risk**	6.524	4.714–9.031	<0.0001

Red text—statistically significant values.

**Table 10 viruses-14-00628-t010:** The total all-cause-death Hazard Ratios for C_2_HEST risk stratification in male cohort.

**Total Death**			
	**HR**	**95%CI**	***p* Value**
**Overall**	1.400	1.331–1.474	<0.0001
**Risk strata**			
**Low risk vs. Medium risk**	3.289	2.559–4.227	<0.0001
**Low risk vs.High risk**	4.476	3.438–5.827	<0.0001

Red text = statistically significant values.

**Table 11 viruses-14-00628-t011:** Associations of individual C_2_HEST score components with mortality in female cohort.

	Component	HR	CI Min.	CI Max.	*p* Value
All-causemortality	Coronaryarterydisease	1.133	0.743	1.728	0.5627
	COPD	2.083	1.299	3.532	0.0064
	Age > 75	2.750	2.088	3.6216	<0.0001
	Thyroiddisease	0.784	0.566	1.105	0.1649
	Hypertension	1.881	1.394	2.537	<0.0001
	HfrEF	1.584	1.134	2.212	0.007

Abbreviations: COPD chronic obstructive pulmonary disease; HfrEF, heart failure with reduce ejection fraction. Red text = statistically significant values.

**Table 12 viruses-14-00628-t012:** Associations of individual C_2_HEST score components with mortality in male cohort.

	Component	HR	CI Min.	CI Max.	*p* Value
All-causemortality	Coronaryarterydisease	1.568	1.180	2.084	0.0019
	COPD	1.182	0.786	1.615	0.4227
	Age > 75	3.0541	2.411	3.869	<0.0001
	Thyroiddisease	1.126	0.688	1.842	0.6378
	Hypertension	1.200	0.952	1.513	0.1233
	HfrEF	1.415	1.055	1.899	0.0206

Abbreviations: COPD, chronic obstructive pulmonary disease; HfrEF, heart failure with reduce ejection fraction. Red text = statistically significant values.

**Table 13 viruses-14-00628-t013:** The log-rank statistics for matching the C_2_HEST risk strata for in-hospital mortality in female cohort.

	H2	h3	h4	h5	h6	h7	h8
m1	164.317	148.669	142.661	121.294	105.396	105.533	10.259
m2		158.373	166.213	158.483	155.603	155.940	12.436
m3			122.464	116.484	116.367	116.190	10.699
m4				79.813	86.505	82.846	8.919
m5					45.423	40.946	6.156
m6						3.820	1.793
m7							0.139

Abbreviations: m, medium; h, high. Red text = statistically significant values.

**Table 14 viruses-14-00628-t014:** The Log-rank statistics for matching the C_2_HEST risk strata for in-hospital mortality in male cohort.

	H2	h3	h4	h5	h6	h7	h8
m1	152.361	134.106	118.904	112.785	98.649	84.149	8.929
m2		152.619	154.813	159.181	155.352	149.997	12.183
m3			116.694	121.473	118.900	115.004	10.673
m4				84.079	82.389	79.865	8.909
m5					58.586	58.244	7.628
m6						32.326	5.686
m7							2.769

Abbreviations: m, medium; h, high. Red text = statistically significant values.

**Table 15 viruses-14-00628-t015:** Clinical non-fatal events in the C_2_HEST risk strata in both study arms.

Variables	Low Risk [0,1]		Medium [2,3]		High Risk [≥4]		*p-*Value OMNIBUS		*p-*Value *f*or Post-Hoc Analysis	
	Females N = 682	Males N = 735	Females N= 284	Males N = 208	Females N = 135	Males N = 139	Females	Males	Females	Males
**Shock** *n*/*n*(%)	34 (5.0)	74 (10.1)	15 (5.3)	31 (14.9)	11 (8.1)	22 (15.8)	0.3314	0.0443	N/A	0.2006 ^a^ 0.1958 ^b^ 1.0 ^c^
**Hypovolemic shock** *n*/*n*(%)	9 (1.3)	13 (1.8)	4 (1.4)	3 (1.4)	5 (3.7)	1 (0.7)	0.1362	0.811	N/A	N/A
**Cardiogenic shock** *n*/*n*(%)	2 (0.3)	5 (0.7)	1 (0.4)	10 (4.8)	5 (3.7)	9 (6.5)	0.0018	<0.0001	1.0 ^a^ 0.0055 ^b^ 0.0439 ^c^	0.0007 ^a^ 0.0002 ^b^ 1.0 ^c^
**Septic shock** *n*/*n*(%)	26 (3.8)	62 (8.4)	12 (4.2)	18 (8.7)	4 (3.0)	18 (12.9)	0.8198	0.2296	N/A	N/A
**Venous thromboembolic disease** *n*/*n*(%)	30 (4.4)	53 (7.2)	18 (6.3)	12 (5.8)	8 (5.9)	7 (5.0)	0.4093	0.5447	N/A	N/A
**Pulmonary embolism** *n*/*n*(%)	24 (3.5)	44 (6.0)	15 (5.3)	11 (5.3)	8 (5.9)	5 (3.6)	0.5516	0.8214	N/A	N/A
**Myocardial infarction** *n*/*n*(%)	2 (0.3)	6 (0.8)	3 (1.1)	7 (3.4)	3 (2.2)	5 (3.6)	0.0251	0.0038	0.464 ^a^ 0.1026 ^b^ 1.0 ^c^	0.035 ^a^ 0.0586 ^b^ 1.0 ^c^
**Myocardial injury, 3x**, *n*/*n*(%)/N	46 (17.5) 263	67 (16.1) 415	51 (29.8) 171	47 (35.1) 134	49 (52.1) 94	38 (39.2) 97	<0.0001	<0.0001	0.0114 ^a^ <0.0001 ^b^ 0.0017 ^c^	<0.0001 ^a,b^ 1.0 ^c^
**Acute heart failure** *n*/*n*(%)	5 (0.7)	3 (0.4)	8 (2.8)	14 (6.7)	24 (17.8)	22 (15.8)	<0.0001	<0.0001	0.0777 ^a^ <0.0001 ^b,c^	<0.0001 ^a,b^ 0.0329 ^c^
**Stroke/TIA** *n*/*n*(%)	4 (0.6)	14 (1.9)	12 (4.2)	7 (3.4)	4 (3.0)	3 (2.2)	0.0002	0.4167	0.0006 ^a^ 0.0872 ^b^ 1.0 ^c^	N/A
**Pneumonia** *n*/*n*(%)	268 (39.3)	414 (56.3)	164 (57.4)	141 (67.8)	88 (65.2)	98 (70.5)	<0.0001	0.0004	<0.0001 ^a, b^ 0.5343 ^c^	0.0117 ^a^ 0.0076 ^b^ 1.0 ^c^
**Complete respiratory failure** *n*/*n*(%)/N	23 (47.9) 48	34 (46.6) 73	16 (43.2) 37	30 (58.8) 51	20 (62.5) 32	23 (65.7) 35	0.2528	0.1348	N/A	N/A
**SIRS** *n*/*n*(%)/N	53 (8.2) 647	89 (12.6) 705	22 (7.8) 283	20 (9.7) 206	21 (15.7) 134	15 (10.8) 139	0.0158	0.4818	1.0 ^a^ 0.0343 ^b^ 0.0636 ^c^	N/A
**Sepsis** *n*/*n*(%)/N	3 (1.0) 288	6 (2.1) 288	3 (2.9) 104	4 (5.1) 79	3 (5.3) 57	4 (5.9) 68	0.053	0.1334	N/A	N/A
**Acute kidney injury** *n*/*n*(%)	37 (5.4)	73 (9.9)	30 (10.6)	37 (17.8)	28 (20.7)	31 (22.3)	<0.0001	<0.0001	0.0193 ^a^ <0.0001 ^b^ 0.0229 ^c^	0.0083 ^a^ 0.0002 ^b^ 1.0 ^c^
**Acute liver dysfunction** *n*/*n*(%)/N	11 (1.9) 592	19 (2.9) 664	12 (4.5) 268	10 (5.1) 197	5 (4.0) 126	9 (7.1) 127	0.0619	0.0458	N/A	0.5214 ^a^ 0.0936 ^b^ 1.0 ^c^
**Multiple organ dysfunction syndrome** *n*/*n*(%)	7 (1.0)	14 (1.9)	3 (1.1)	5 (2.4)	4 (3.0)	4 (2.9)	0.1674	0.6162	N/A	N/A
**Bleedings** *n*/*n*(%)	27 (4.0)	37 (5.0)	13 (4.6)	12 (5.8)	9 (6.7)	16 (11.5)	0.3758	0.0128	N/A	1.0 ^a^ 0.0184 ^b^ 0.2545 ^c^

Continuous variables are presented as: mean ± SD range (minimum-maximum) and number of non-missing values. Categorized variables are presented as: a number with a percentage. Abbreviations: N, valid measurements; *n*, number of patients with parameter above cut-off point; SD, standard deviation; OMNIBUS, analysis of variance; TIA, transient ischemic attack; SIRS, systemic inflammatory response syndrome; N/A, non-applicable; ^a^ low risk vs. medium risk, ^b^ low risk vs. high risk, ^c^ medium risk vs. high risk. Red color text = statistically significant values.

## Data Availability

The datasets used and/or analyzed during the current study are available from the corresponding author upon reasonable request.

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
