# Peer review of "Sex-Dependent Differences in Predictive Value of the C2HEST Score in Subjects with COVID-19—A Secondary Analysis of the COLOS Study"

_viruses, 2022, doi:10.3390/v14030628_

Round 1

Reviewer 1 Report

The manuscript entitled “Sex-dependent differences in predictive value of C2HEST score in subjects with COVID-19 – a secondary analysis of the 3 COLOS study” by Rola et al. describes the results of C2HEST score, which predict the mortality with highest sensitivity in female population during hospitalization, three- and six-months follow-up. The authors raise a clinically interesting predictive risk sex-dependent value of the C2HEST score in patients admitted to hospital due to SARS-CoV-2 infection.

However, the limitation of the study is single-centre retrospective analysis.

Comments to authors:
Overall, the manuscript theme is interesting and valuable for clinic; however, this reviewer has a few minor comments to the manuscript.

  1. The figure 3 is lacking time unit.
  2. The authors should describe/explain the results of figure 4, as there are differences between female and male cohorts.
  3. The authors should explain more in text the results of tables 12 and 13. In addition, the tables need better presentation/explanation for the reader.
  4. The authors should elaborate more about bleedings.
  5. The manuscript should be checked for any word clumps.

Author Response

We would like to thank the Reviewer for an in-depth analysis of the manuscript and for pivotal comments provided, which have resulted in a significant improvement of this manuscript.

Minor comments

Ad 1. We would like to thank the Reviewer for finding missing data in figure 1. – we have fixed it, according to the Reviewer’s suggestion.

Ad 2. We believe that the Reviewer made a valid point when suggested adding a short description of the Figure 4. As result, we have added in results section the following:

 “We have observed differences in estimated survival probability in both study cohorts. Practically, starting from admission time, the females were more likely to survive the COVID-19. Estimated six-month survival probability for high-risk subjects reached 0.5 in the female cohort, while for the male subject was below 0.4. Similarly, in medium-risk-stratum for women the survival probability was above 0.6 when compared to 0.5 in men. Additionally, the low-risk subjects in the female cohort maintained at the level of more than 0.9 for the whole observation period while in men reached 0.8, respectively.”

Ad 3.  We agree with the Reviewer, that the clarification of the log-rank analysis was not precise. This analysis was conducted in order to verify if the original cut-off point values for particular risk strata in the C2HEST score, used for the original purpose, fit to the currently assessed mortality-analysis in both genders. To verify, if any other cut-off point values could better distinguish the strata regarding mortality, such analysis was performed for women and men separately. We have commented on it in the Results, according to the Reviewer’s suggestion as following:

Additionally, we verified, whether the original cut-off values for particular C2HEST score risk (the low/medium/high-risk categories for 0-1/2-3/4 points, respectively) are potentially the best possible stratification system. Regarding the difference in Kaplan-Meier survival curves, all the possible C2HEST intervals were analyzed in both study cohorts, and for each, we calculated the log-rank statistics (Table 12 and 13). The highest value of log-rank test statistics, presenting the best cut-off point for high (h) and medium (m) strata was obtained for the original C2HEST-score risk strata in the female population (m2 and h4, respectively). On the other hand, in the male cohort, the highest value of the Log-rank corresponded with m2 and h5, which reflects the following strata: 0-1 low, 2-4 medium, 5-8 high.

Ad 4. We would like to thank the Reviewer for a suggestion regarding the impact of bleeding issues on outcome of COVID, we are pleased to inform we added short comments in the Discussion section:

“Although the principal clinical manifestation of severe COVID-19 is a respiratory failure with a coexisting uncontrolled immune reaction, subjects with COVID-19 show a high incidence of thromboembolic events [31], particularly in fatal cases [32], however antithrombotic treatment prior to COVID-19 infection is unlikely to have a protective effect [33]. Bleeding complications in subjects with COVID-19 give rise to justifiable concerns [34,35] and should always be considered before applying anticoagulation in patients with SARS-CoV-2 infection. Therefore, several predictive scores focused on identifying patients at increased risk for major bleeding have been recently proposed. Results of our study suggest that the C2HEST score might be also useful in the identification of the “high-risk for bleeding” subpopulations. However, subsequent studies are needed to define the predictive value of the C2HEST score in terms of bleedings.

Ad.5 We have revised manuscript and removed all the noticed word clumps.

Reviewer 2 Report

The topic of this manuscript falls within the scope of Viruses Journal. The purpose of the study was to assess the sex-dependent predictive value of C2HEST-scores. The Authors showed e better C2HEST-score predictive value for mortality in women and illustrated sex-dependent differences predicting the non-fatal secondary outcomes.

Authors used appropriate statistic methods. The conclusions are consistent with presented evidence and arguments.

the strength of this paper: very interesting topic, introduction-relevant and concise; material and methods-the right choice of methodology methods, which was presented in comprehensible way; the obtained results are presented in the form of figures, which are clear and easy to understand; the discussion- supports the results properly and refers to the current literature in appropriate manner; the conclusions- based on the obtained results; very good limitations of the study

There are some comments in the reviewer opinion which should be taken under consideration by the Authors:

  1. In the intorduction please give current data on COVID infection [WHO Coronavirus (COVID-19) Dashboard. . Available online: https://covid19.who.int/
  2. Please give citations to the text line 81-82.
  3. in the discussion/limitation, please refer to the issue:

Based on some reports, it seems that individuals with gastrointestinal problems are more likely to experience severe COVID-19 disease, which may be seen as a predictor of the development of severe respiratory disorders. The hepatic consequences of SARS-CoV-2 infection are an important problematic component of COVID-19 that is most important in patients with earlier liver disease who are at remarkably high risk of severe COVID-19 and death (PMID: 34768321, PMID: 32717345)

Author Response

We would like to thank the Reviewer for an in-depth analysis of the manuscript and for pivotal comments provided, which have resulted in a significant improvement of this manuscript.

Introduction section:

Ad 1 As the Reviewer suggested, we have made a comment on the current data on COVID infection necessary references have been added.

Ad 2. We believe that the Reviewer made a valid point when suggested adding missing references in line 81-82.

Discussion section:

Ad 3 Another valid point mentioned by the Reviewer is a remark regarding, impact of GI and liver diseases on outcomes of COVID-diseases therefore we added short comment key on this topic in discussion section. “Furthermore, some data [25,26] suggest that individuals with gastrointestinal problems particularly those with earlier stages of liver impairment are more prone to develop severe COVID-19 disease with advanced respiratory failure. Concerning epidemiological data suggesting a higher prevalence of liver disorders [27] with coexisting higher susceptibility for endothelial dysfunction [28,29] may be important factors affecting outcomes in the male population.